# THE RISKS OF INVARIANT RISK MINIMIZATION

**Elan Rosenfeld, Pradeep Ravikumar, Andrej Risteski**
Machine Learning Department
Carnegie Mellon University
`elan@cmu.edu, pradeepr@cs.cmu.edu, aristesk@andrew.cmu.edu`

## ABSTRACT

Invariant Causal Prediction (Peters et al., 2016) is a technique for out-of-distribution generalization which assumes that some aspects of the data distribution vary across the training set but that the underlying causal mechanisms remain constant. Recently, Arjovsky et al. (2019) proposed Invariant Risk Minimization (IRM), an objective based on this idea for learning deep, invariant features of data which are a complex function of latent variables; many alternatives have subsequently been suggested. However, formal guarantees for all of these works are severely lacking. In this paper, we present the first analysis of classification under the IRM objective—as well as these recently proposed alternatives—under a fairly natural and general model. In the linear case, we give simple conditions under which the optimal solution succeeds or, more often, fails to recover the optimal invariant predictor. We furthermore present the *very first results in the non-linear regime*: we demonstrate that IRM can fail catastrophically unless the test data are sufficiently similar to the training distribution—this is precisely the issue that it was intended to solve. Thus, in this setting we find that IRM and its alternatives fundamentally *do not improve* over standard Empirical Risk Minimization.

## 1 INTRODUCTION

Prediction algorithms are evaluated by their performance on unseen test data. In classical machine learning, it is common to assume that such data are drawn i.i.d. from the same distribution as the data set on which the learning algorithm was trained—in the real world, however, this is often not the case. When this discrepancy occurs, algorithms with strong in-distribution generalization guarantees, such as Empirical Risk Minimization (ERM), can fail catastrophically. In particular, while deep neural networks achieve superhuman performance on many tasks, there is evidence that they rely on statistically informative but non-causal features in the data (Beery et al., 2018; Geirhos et al., 2018; Ilyas et al., 2019). As a result, such models are prone to errors under surprisingly minor distribution shift (Su et al., 2019; Recht et al., 2019). To address this, researchers have investigated alternative objectives for training predictors which are robust to possibly egregious shifts in the test distribution.

The task of generalizing under such shifts, known as *Out-of-Distribution (OOD) Generalization*, has led to many separate threads of research. One approach is Bayesian deep learning, accounting for a classifier's uncertainty at test time (Neal, 2012). Another technique that has shown promise is data augmentation—this includes both automated data modifications which help prevent overfitting (Shorten & Khoshgoftaar, 2019) and specific counterfactual augmentations to ensure invariance in the resulting features (Volpi et al., 2018; Kaushik et al., 2020).

A strategy which has recently gained particular traction is Invariant Causal Prediction (ICP; Peters et al. 2016), which views the task of OOD generalization through the lens of causality. This framework assumes that the data are generated according to a Structural Equation Model (SEM; Bollen 2005), which consists of a set of so-called mechanisms or structural equations that specify variables given their parents. ICP assumes moreover that the data can be partitioned into *environments*, where each environment corresponds to interventions on the SEM (Pearl, 2009), but where the mechanism by which the target variable is generated via its direct parents is unaffected. Thus the causal mechanism of the target variable is unchanging but other aspects of the distribution can vary broadly. As a result, learning mechanisms that are the same across environments ensures recovery of the invariant features which generalize under arbitrary interventions. In this work, we consider objectives that attempt to

learn what we refer to as the "optimal invariant predictor"—this is the classifier which uses and is optimal with respect to only the invariant features in the SEM. By definition, such a classifier does not overfit to environment-specific properties of the data distribution, so it will generalize even under major distribution shift at test time. In particular, we focus our analysis on one of the more popular objectives, Invariant Risk Minimization (IRM; Arjovsky et al. (2019)), but our results can easily be extended to similar recently proposed alternatives.

Various works on invariant prediction (Muandet et al., 2013; Ghassami et al., 2017; Heinze-Deml et al., 2018; Rojas-Carulla et al., 2018; Subbaswamy et al., 2019; Christiansen et al., 2020) consider regression in both the linear and non-linear setting, but they exclusively focus on learning with fully or partially observed covariates or some other source of information. Under such a condition, results from causal inference (Maathuis et al., 2009; Peters et al., 2017) allow for formal guarantees of the identification of the invariant features, or at least a strict subset of them. With the rise of deep learning, more recent literature has developed objectives for learning invariant representations when the data are a non-linear function of unobserved latent factors, a common assumption when working with complex, high-dimensional data such as images. Causal discovery and inference with unobserved confounders or latents is a much harder problem (Peters et al., 2017), so while empirical results seem encouraging, these objectives are presented with few formal guarantees. IRM is one such objective for invariant representation learning. The goal of IRM is to learn a feature embedder such that the optimal linear predictor on top of these features is the same for every environment—the idea being that only the invariant features will have an optimal predictor that is invariant. Recent works have pointed to shortcomings of IRM and have suggested modifications which they claim prevent these failures. However, these alternatives are compared in broad strokes, with little in the way of theory.

In this work, we present the first formal analysis of classification under the IRM objective under a fairly natural and general model which carefully formalizes the intuition behind the original work. Our results show that despite being inspired by invariant prediction, this objective can frequently be expected to perform *no better than ERM*. In the linear setting, we present simple, exact conditions under which solving to optimality succeeds or, more often, breaks down in recovering the optimal invariant predictor. We also demonstrate another major failure case—under mild conditions, there exists a feasible point that uses only non-invariant features and achieves lower empirical risk than the optimal invariant predictor; thus it will appear as a more attractive solution, yet its reliance on non-invariant features mean it will fail to generalize. As corollaries, we present similar settings where all recently suggested alternatives to IRM likewise fail. Futhermore, we present the *first results in the non-linear regime*: we demonstrate the existence of a classifier with exponentially small sub-optimality which nevertheless heavily relies on non-invariant features on most test inputs, resulting in worse-than-chance performance on distributions that are sufficiently dissimilar from the training environments. These findings strongly suggest that existing approaches to ICP for high-dimensional latent variable models do not cleanly achieve their stated objective and that future work would benefit from a more formal treatment.

## 2 Related work

Works on learning deep invariant representations vary considerably: some search for a *domain-invariant* representation (Muandet et al., 2013; Ganin et al., 2016), i.e. invariance of the distribution $p(\Phi(x))$, typically used for domain adaptation (Ben-David et al., 2010; Ganin & Lempitsky, 2015; Zhang et al., 2015; Long et al., 2018), with assumed access to labeled or unlabeled data from the target distribution. Other works instead hope to find representations that are *conditionally* domain-invariant, with invariance of $p(\Phi(x) \mid y)$ (Gong et al., 2016; Li et al., 2018). However, there is evidence that invariance may not be sufficient for domain adaptation (Zhao et al., 2019; Johansson et al., 2019). In contrast, this paper focuses instead on *domain generalization* (Blanchard et al., 2011; Rosenfeld et al., 2021), where access to the test distribution is not assumed.

Recent works on domain generalization, including the objectives discussed in this paper, suggest invariance of the *feature-conditioned label distribution*. In particular, Arjovsky et al. (2019) only assume invariance of $\mathbb{E}[y \mid \Phi(x)]$; follow-up works rely on a stronger assumption of invariance of higher conditional moments (Krueger et al., 2020; Xie et al., 2020; Jin et al., 2020; Mahajan et al., 2020; Bellot & van der Schaar, 2020). Though this approach has become popular in the last year, it is somewhat similar to the existing concept of *covariate shift* (Shimodaira, 2000; Bickel et al., 2009),

which considers the same setting. The main difference is that these more recent works assume that the shifts in $p(\Phi(x))$ occur between discrete, labeled environments, as opposed to more generally from train to test distributions.

Some concurrent lines of work study different settings yet give results which are remarkably similar to ours. Xu et al. (2021) show that an infinitely wide two-layer network extrapolates linear functions when the training data is sufficiently diverse. In the context of domain generalization specifically, Rosenfeld et al. (2021) prove that ERM remains optimal for both interpolation and extrapolation in the linear setting and that the latter is exponentially harder than the former. These results mirror our findings that none of the studied objectives outperform ERM.

## 3 MODEL AND INFORMAL RESULTS

We consider an SEM with explicit separation of *invariant* features $z_c$, whose joint distribution with the label is fixed for all environments, and *environmental* features $z_e$ ("non-invariant"), whose distribution can vary. This choice is to ensure that our model properly formalizes the intuition behind invariant prediction techniques such as IRM, whose objective is to ensure generalizing predictors by recovering only the invariant features—we put off a detailed description of these objectives until after we have introduced the necessary terminology.

We assume that data are drawn from a set of $E$ training environments $\mathcal{E} = \{e_1, e_2, \ldots, e_E\}$ and that we know from which environment each sample is drawn. For a given environment $e$, the data are defined by the following process: first, a label $y \in \{\pm 1\}$ is drawn according to a fixed probability:

$$y = \begin{cases} 1, & \text{w.p. } \eta \\ -1, & \text{otherwise.} \end{cases} \tag{1}$$

Next, both invariant features and environmental features are drawn according to a Gaussian:[1]

$$z_c \sim \mathcal{N}(y \cdot \mu_c, \sigma_c^2 I), \qquad z_e \sim \mathcal{N}(y \cdot \mu_e, \sigma_e^2 I), \tag{2}$$

with $\mu_c \in \mathbb{R}^{d_c}, \mu_e \in \mathbb{R}^{d_e}$—typically, for complex, high-dimensional data we would expect $E < d_c \ll d_e$. Finally, the observation $x$ is generated as a function of the latent features:

$$x = f(z_c, z_e). \tag{3}$$

The complete data generating process is displayed in Figure 3.1. We assume $f$ is injective, so that it is in principle possible to recover the latent features from the observations, i.e. there exists a function $\Phi$ such that $\Phi(f(z_c, z_e)) = [z_c, z_e]^T$. We remark that this our *only* assumption on $f$, even when it is non-linear. Further, note that we model class-conditional means as direct opposites merely for clarity, as it greatly simplifies the calculations. None of our proofs require this condition: it is straightforward to extend our results to arbitrary means, and the non-linear setting also allows for arbitrary covariances. In fact, our proof technique for non-linear $f$ could be applied to *any* distribution that sufficiently concentrates about its mean (e.g., sub-Gaussian). We write the joint and marginal distributions as $p^e(x, y, z_c, z_e)$. When clear from context, we omit the specific arguments.

**Remarks on the model.** This model is natural and flexible; it generalizes several existing models used to analyze learning under the existence of adversarial distribution shift or non-invariant correlations (Schmidt et al., 2018; Sagawa et al., 2020). The fundamental facet of this model is the constancy of the invariant parameters $\eta, \mu_c, \sigma_c^2, f$ across environments—the dependence of $\mu_e, \sigma_e$ on the environment allows for varying distributions, while the true causal process remains unchanged. Here we make a few clarifying remarks:

- We do not impose any constraints on the model parameters. In particular, we do not assume a prior over the environmental parameters. Observe that $\mu_c, \sigma_c^2$ are the same for all environments,

---

[1]Note the deliberate choice to have $z_e$ depend on $y$. Much work on this problem models spurious features which correlate with the label but are not causal. However, the term "spurious" is often applied incongruously; in recent work, the term has been co-opted to refer to any feature that correlates with the label but does not cause it. Thus there is a subtle distinction: if we allow for anti-causality, i.e. the label causing the features, the resulting correlation is *not* spurious. We therefore avoid using the term "spurious" in this work.

hence the subscript indicates the invariant relationship. In contrast, with some abuse of notation, the environmental subscript is used to indicate both dependence on the environment and the index of the environment itself (e.g., $\mu_i$ represents the mean specific to environment $i$).

- While we have framed the model as $y$ causing $z_c$, the causation can just as easily be viewed in the other direction. The log-odds of $y$ are a linear function of $z_c$—this matches logistic regression with an invariant regression vector $\beta_c = 2\mu_c/\sigma_c^2$ and bias $\beta_0 = \log \frac{\eta}{1-\eta}$. We present the model as above to emphasize that the causal relationships between $y$ and the $z_c, z_e$ are a priori indistinguishable, and because we believe this direction is more intuitive.

We consider the setting where we are given infinite samples from each environment; this allows us to isolate the behavior of the objectives themselves, rather than finite-sample effects. Upon observing samples from this model, our objective is thus to learn a feature embedder $\Phi$ and classifier[2] $\hat{\beta}$ to minimize the risk on an unseen environment $e$:

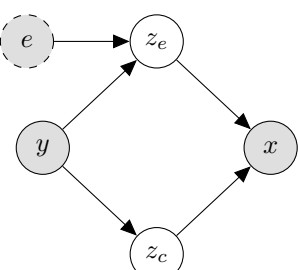

$$\mathcal{R}^e(\Phi, \hat{\beta}) := \mathbb{E}_{(x,y)\sim p^e}\left[\ell(\sigma(\hat{\beta}^T\Phi(x)), y)\right].$$

The function $\ell$ can be any loss appropriate to classification: in this work we consider the logistic and the 0-1 loss. Note that we are *not* hoping to minimize risk in expectation over the environments; this is already accomplished via ERM or distributionally robust optimization (DRO; Bagnell 2005; Ben-Tal et al. 2009). Rather, we hope to extract and regress on invariant features while ignoring environmental features, such that our predictor generalizes to all

Figure 3.1: A Bayesian network depicting our model. Shading indicates the variable is observed.

unseen environments regardless of their parameters. In other words, the focus is on minimizing risk in the worst-case. We refer to the predictor which will minimize worst-case risk under arbitrary distribution shift as the *optimal invariant predictor*. To discuss this formally, we define precisely what we mean by this term.

**Definition 1.** Under the model described by Equations 1-3, the *optimal invariant predictor* is the predictor defined by the composition of a) the featurizer which recovers the invariant features and b) the classifier which is optimal with respect to those features:

$$\Phi^*(x) := \begin{bmatrix} I & 0 \\ 0 & 0 \end{bmatrix} \circ f^{-1}(x) = [z_c], \quad \hat{\beta}^* := \begin{bmatrix} \beta_c \\ \beta_0 \end{bmatrix} := \begin{bmatrix} 2\mu_c/\sigma_c^2 \\ \log\frac{\eta}{1-\eta} \end{bmatrix}.$$

Observe that this definition closely resembles Definition 3 of Arjovsky et al. (2019); the only difference is that here the optimal invariant predictor must recover *all invariant features*. As Arjovsky et al. (2019) do not posit a data model, the concept of recovering "all invariant features" is not well-defined for their setting; technically, a featurizer which outputs the empty set would elicit an invariant predictor, but this would not satisfy the above definition. The classifier $\hat{\beta}^*$ is optimal with respect to the invariant features and so it achieves the minimum possible risk without using environmental features. Observe that *the optimal invariant predictor is distinct from the Bayes classifier*; the Bayes classifier uses environmental features which are informative of the label but non-invariant; the optimal invariant predictor *explicitly ignores* these features.

With the model defined, we can informally present our results; we defer the formal statements to first give a background on the IRM objective in the next section. With a slight abuse of notation, we identify a predictor by the tuple $\Phi, \hat{\beta}$ which parametrizes it. First, we show that the usefulness of IRM exhibits a "thresholding" behavior depending on $E$ and $d_e$:

**Theorem 3.1** (Informal, Linear). *For linear $f$, consider solving the IRM objective to learn a linear $\Phi$ with invariant optimal classifier $\hat{\beta}$. If $E > d_e$, then $\Phi, \hat{\beta}$ is precisely the optimal invariant predictor; it uses only invariant features and generalizes to all environments with minimax-optimal risk. If $E \leq d_e$, then $\Phi, \hat{\beta}$ relies upon non-invariant features.*

In fact, when $E \leq d_e$ it is even possible to learn a classifier *solely* relying on environmental features that achieves lower risk on the training environments than the optimal invariant predictor:

---

[2]Following the terminology of Arjovsky et al. (2019), we refer to the regression vector $\hat{\beta}$ as a "classifier" and the composition of $\Phi, \hat{\beta}$ as a "predictor".

**Theorem 3.2** (Informal, Linear). *For linear $f$ and $E \leq d_e$ there exists a linear predictor $\Phi, \hat{\beta}$ which* uses *only environmental features, yet achieves lower risk than the optimal invariant predictor.*

Finally, in the non-linear case, we show that IRM fails unless the training environments approximately "cover" the space of possible environments, and therefore it behaves similarly to ERM:

**Theorem 3.3** (Informal, Non-linear). *For arbitrary $f$, there exists a non-linear predictor $\Phi, \hat{\beta}$ which is nearly optimal under the penalized objective and furthermore is nearly identical to the optimal invariant predictor on the training distribution. However, for any test environment with a mean sufficiently different from the training means, this predictor will be equivalent to the ERM solution on nearly all test points. For test distributions where the environmental feature correlations with the label are reversed, this predictor has almost 0 accuracy.*

**Extensions to other objectives.** Many follow-up works have suggested alternatives to IRM—some are described in the next section. Though these objectives perform better on various baselines, there are few formal guarantees and no results beyond the linear case. Due to their collective similarities, we can easily derive corollaries which extend every theorem in this paper to these objectives, demonstrating that they all suffer from the same shortcomings. Appendix E contains example corollaries for each of the results presented in this work.

## 4 BACKGROUND ON IRM AND ITS ALTERNATIVES

During training, a classifier will learn to leverage correlations between features and labels in the training data to make its predictions. If a correlation varies with the environment, it may not be present in future test distributions—worse yet, it may be *reversed*—harming the classifier's predictive ability. IRM (Arjovsky et al., 2019) is a recently proposed approach to learning environmentally invariant representations to facilitate invariant prediction.

**The IRM objective.** IRM posits the existence of a feature embedder $\Phi$ such that the optimal classifier on top of these features is the same for every environment. The authors argue that such a function will use only invariant features, since non-invariant features will have different joint distributions with the label and therefore a fixed classifier on top of them won't be optimal in all environments. To learn this $\Phi$, the IRM objective is the following constrained optimization problem:

$$\min_{\Phi, \hat{\beta}} \quad \frac{1}{|\mathcal{E}|} \sum_{e \in \mathcal{E}} \mathcal{R}^e(\Phi, \hat{\beta}) \qquad \text{s.t.} \quad \hat{\beta} \in \arg\min_{\beta} \mathcal{R}^e(\Phi, \beta) \quad \forall e \in \mathcal{E}. \tag{4}$$

This bilevel program is highly non-convex and difficult to solve. To find an approximate solution, the authors consider a Langrangian form, whereby the sub-optimality with respect to the constraint is expressed as the squared norm of the gradients of each of the inner optimization problems:

$$\min_{\Phi, \hat{\beta}} \quad \frac{1}{|\mathcal{E}|} \sum_{e \in \mathcal{E}} \left[ \mathcal{R}^e(\Phi, \hat{\beta}) + \lambda \|\nabla_{\hat{\beta}} \mathcal{R}^e(\Phi, \hat{\beta})\|_2^2 \right]. \tag{5}$$

Assuming the inner optimization problem is convex, achieving feasibility is equivalent to the penalty term being equal to 0. Thus, Equations 4 and 5 are equivalent if we set $\lambda = \infty$.

**Alternative objectives.** IRM is motivated by the existence of a featurizer $\Phi$ such that $\mathbb{E}[y \mid \Phi(x)]$ is invariant. Follow-up works have proposed variations on this objective, based instead on the strictly stronger desideratum of the invariance of $p(y \mid \Phi(x))$. Krueger et al. (2020) suggest penalizing the variance of the risks, while Xie et al. (2020) give the same objective but taking the square root of the variance. Many papers have suggested similar alternatives (Jin et al., 2020; Mahajan et al., 2020; Bellot & van der Schaar, 2020). These objectives are compelling—indeed, it is easy to show that the optimal invariant predictor constitutes a stationary point of each of these objectives:

**Proposition 4.1.** *Suppose the observed data are generated according to Equations 1-3. Then the (parametrized) optimal invariant predictor $\Phi^*, \hat{\beta}^*$ is a stationary point for Equation 4.*

The stationarity of the optimal invariant predictor for the other objectives is a trivial corollary. However, in the following sections we will demonstrate that such a result is misleading and that a more careful investigation is necessary.

## 5 THE DIFFICULTIES OF IRM IN THE LINEAR REGIME

In their work proposing IRM, Arjovsky et al. (2019) present specific conditions for an upper bound on the number of training environments needed such that a feasible linear featurizer $\Phi$ will have an invariant optimal regression vector $\hat{\beta}$. Our first result is similar in spirit but presents a substantially stronger (and simplified) upper bound in the classification setting, along with a matching lower bound: we demonstrate that observing a large number of environments—linear in the number of environmental features—is *necessary* for generalization in the linear regime.

**Theorem 5.1** (Linear case). *Assume $f$ is linear. Suppose we observe $E$ training environments. Then the following hold:*

1. *Suppose $E > d_e$. Consider any linear featurizer $\Phi$ which is feasible under the IRM objective (4), with invariant optimal classifier $\hat{\beta} \neq 0$, and write $\Phi(f(z_c, z_e)) = Az_c + Bz_e$. Then under mild non-degeneracy conditions, it holds that $B = 0$. Consequently, $\hat{\beta}$ is the optimal classifier for all possible environments.*

2. *If $E \leq d_e$ and the environmental means $\mu_e$ are linearly independent, then there exists a linear $\Phi$—where $\Phi(f(z_c, z_e)) = Az_c + Bz_e$ with $rank(B) = d_e + 1 - E$—which is feasible under the IRM objective. Further, both the logistic and 0-1 risks of this $\Phi$ and its corresponding optimal $\hat{\beta}$ are strictly lower than those of the optimal invariant predictor.*

Similar to Arjovsky et al. (2019), the set of environments which do not satisfy Theorem 5.1 has measure zero under any absolutely continuous density over environmental parameters. Further details, and the full proof, can be found in Appendix C.1. Since the optimal invariant predictor is Bayes with respect to the invariant features, by the data-processing inequality the only way a predictor can achieve lower risk is by relying on environmental features. Thus, Theorem 5.1 directly implies that when $E \leq d_e$, the global minimum necessarily uses these non-invariant features and therefore will not universally generalize to unseen environments. On the other hand, in the (perhaps unlikely) case that $E > d_e$, any feasible solution will generalize, and the optimal invariant predictor has the minimum (and minimax) risk of all such predictors:

**Corollary 5.2.** *For both logistic and 0-1 loss, the optimal invariant predictor is the global minimum of the IRM objective if and only if $E > d_e$.*

Let us compare our theoretical findings to those of Arjovsky et al. (2019). Suppose the observations $x$ lie in $\mathbb{R}^d$. Roughly, their theorem says that for a learned $\Phi$ of rank $r$ with invariant optimal coefficient $\hat{\beta}$, if the training set contains $d - r + d/r$ "non-degenerate" environments, then $\hat{\beta}$ will be optimal for all environments. There are several important issues with this result: first, they present no result tying the rank of $\Phi$ to their actual objective; their theory thus motivates the objective, but does not provide any performance guarantees for its solution. Next, observe when $x$ is high-dimensional (i.e. $d \gg d_e + d_c$—in which case $\Phi$ will be comparatively low-rank (i.e. $r \leq d_e + d_c$)—their result requires $\Omega(d)$ environments, which is extreme. For example, think of images on a low-dimensional manifold embedded in very high-dimensional space. Even when $d = d_c + d_e$, the "ideal" $\Phi$ which recovers precisely $z_c$ would have rank $d_c$, and therefore their condition for invariance would require $E > d_e + d_e/d_c$, a stronger requirement than ours; this inequality also seems unlikely to hold in most real-world settings. Finally, they give no lower bound on the number of required environments—prior to this work, there were no existing results for the performance of the IRM objective when their conditions are not met. We also run a simple synthetic experiment to verify our theoretical results, drawing samples according to our model and learning a predictor with the IRM objective. Details and results of this experiment can be found in Appendix C.2. We now sketch a constructive proof of part 2 of the theorem for when $E = d_e$:

*Proof Sketch.* Since $f$ has an inverse over its range, we can define $\Phi$ as a linear function directly over the latents $[z_c, z_e]$. Specifically, define $\Phi(x) = [z_c, p^T z_e]$. Here, $p$ is a unit-norm vector such that $\forall e \in \mathcal{E}, \ p^T \mu_e = \sigma_e^2 \tilde{\mu}$; $\tilde{\mu}$ is a fixed scalar that depends on the geometry of $\mu_e, \sigma_e^2$—such a vector exists so long as the means are linearly independent. Observe that this $\Phi$ also has the desired rank. Since this is a linear function of a multivariate Gaussian, the label-conditional distribution of each environment's non-invariant latents has a simple closed form: $p^T z_e \mid y \sim \mathcal{N}(y \cdot p^T \mu_e, \|p\|_2^2 \sigma_e^2) \stackrel{d}{=} \mathcal{N}(y \cdot \sigma_e^2 \tilde{\mu}, \sigma_e^2)$.

For separating two Gaussians, the optimal linear classifier is $\Sigma^{-1}(\mu_1 - \mu_0)$—here, the optimal classifier on $p^T z_e$ is precisely $2\tilde{\mu}$, which does not depend on the environment (and neither do the optimal coefficients for $z_c$). Though the distribution varies across environments, the optimal classifier is the same! Thus, $\Phi$ directly depends on the environmental features, yet the optimal regression vector $\hat{\beta}$ for each environment is constant. To see that it has lower risk than the optimal invariant predictor, note that this classifier is Bayes with respect to its features and that the optimal invariant predictor uses a strict subset of these features, and therefore it has less information for its predictions. □

**A purely environmental predictor.** The precise value of $\tilde{\mu}$ in the proof sketch above represents how strongly this non-invariant feature is correlated with the label. In theory, a predictor that achieves a lower objective value could do so by a very small margin—incorporating an arbitrarily small amount of information from a non-invariant feature would suffice. This result would be less surprising, since achieving low empirical risk might still ensure that we are "close" to the optimal invariant predictor. Our next result shows that this is not the case: there exists a feasible solution which uses *only the environmental features* yet performs better than the optimal invariant predictor on all $e \in \mathcal{E}$ for which $\tilde{\mu}$ is large enough.

**Theorem 5.3.** *Suppose we observe $E \leq d_e$ environments, such that all environmental means are linearly independent. Then there exists a feasible $\Phi, \hat{\beta}$ which uses* only *environmental features and achieves lower 0-1 risk than the optimal invariant predictor on every environment $e$ such that $\sigma_e \tilde{\mu} > \sigma_c^{-1}\|\mu_c\|_2$ and $2\sigma_e \tilde{\mu}\sigma_c^{-1}\|\mu_c\|_2 \geq |\beta_0|$.*

The latter of these two conditions is effectively trivial, requiring only a small separation of the means and balance in class labels. From the construction of $\tilde{\mu}$ in the proof of Lemma C.1, we can see that the former condition is more likely to be met when $E \ll d_e$ and in environments where some non-invariant features are reasonably correlated with the label—both of which can be expected to hold in the high-dimensional setting. Figure C.2 in the appendix plots the results for a few toy examples for various dimensionalities and variances to see how often this condition holds in practice. For all settings, the number of environments observed before the condition ceases to hold is quite high—on the order of $d_e - d_c$.

## 6 THE FAILURE OF IRM IN THE NON-LINEAR REGIME

We've demonstrated that OOD generalization is difficult in the linear case, but it is achievable given enough training environments. Our results—and those of Arjovsky et al. (2019)—intuitively proceed by observing that each environment reduces a "degree of freedom" of the solution, such that only the invariant features remain feasible if enough environments are seen. In the non-linear case, it's not clear how to capture this idea of restricting the "degrees of freedom"—and in fact our results imply that this intuition is simply wrong. Instead, we show that the solution generalizes only to test environments that are sufficiently similar to the training environments. Thus, these objectives present no real improvement over ERM or DRO.

Non-linear transformations of the latent variables make it hard to characterize the optimal linear classifier, which makes reasoning about the constrained solution to Equation 4 difficult. Instead we turn our attention to Equation 5, the penalized IRM objective. In this section we demonstrate a foundational flaw of IRM in the non-linear regime: unless we observe enough environments to "cover" the space of non-invariant features, a solution that appears to be invariant can still wildly underperform on a new test distribution. We begin with a definition about the optimality of a coefficient vector $\hat{\beta}$:

**Definition 2.** For $0 < \gamma < 1$, a coefficient vector $\hat{\beta}$ is $\gamma$-*close to optimal* for a label-conditional feature distribution $z \sim \mathcal{N}(y \cdot \mu, \Sigma)$ if

$$\hat{\beta}^T \mu \geq (1 - \gamma)2\mu^T \Sigma^{-1} \mu.$$

Since the optimal coefficient vector is precisely $2\Sigma^{-1}\mu$, being $\gamma$-close implies that $\hat{\beta}$ is reasonably aligned with that optimum. Observe that the definition does not account for magnitude—the set of vectors which is $\gamma$-close to optimal is therefore a halfspace which is normal to the optimal vector. One of our results in the non-linear case uses the following assumption, which says that the observed environmental means are sufficiently similar to one another.

**Assumption 1.** There exists a $0 \leq \gamma < 1$ such that the ERM-optimal classifier for the non-invariant features,

$$\beta_{e;\text{ERM}} := \arg\min_{\hat{\beta}_e} \frac{1}{|\mathcal{E}|} \sum_{e \in \mathcal{E}} \mathbb{E}_{z_c, z_e, y \sim p^e} \left[ \ell(\sigma(\beta_c^T z_c + \hat{\beta}_e^T z_e + \beta_0), y) \right], \tag{6}$$

is $\gamma$-close to optimal for every environmental feature distribution in $\mathcal{E}$.

This assumption says the environmental distributions are similar enough such that the optimal "average classifier" is reasonably predictive for each environment individually. This is a natural expectation: we are employing IRM precisely *because* we expect the ERM classifier to do well on the training set but fail to generalize. If the environmental parameters are sufficiently orthogonal, we might instead expect ERM to ignore the features which are not at least moderately predictive across all environments. Finally, we note that if this assumption only holds for a subset of features, our result still applies by marginalizing out the dimensions for which it does not hold.

We are now ready to give our main result in the non-linear regime. We present a simplified version, assuming that that $\sigma_e^2 = 1 \; \forall e$. This is purely for clarity of presentation; the full theorem is presented in Appendix D. We make use of two constants in the following proof—the average squared norm of the environmental means, $\overline{\|\mu\|_2^2} := \frac{1}{E} \sum_{e \in \mathcal{E}} \|\mu_e\|_2^2$; and the standard deviation of the response variable of the ERM-optimal classifier, $\sigma_{\text{ERM}} := \sqrt{\|\beta_c\|_2^2 \sigma_c^2 + \|\beta_{e;\text{ERM}}\|_2^2 \sigma_e^2}$.

**Theorem 6.1** (Non-linear case, simplified). *Suppose we observe $E$ environments $\mathcal{E} = \{e_1, e_2, \ldots, e_E\}$, where $\sigma_e^2 = 1 \; \forall e$. Then, for any $\epsilon > 1$, there exists a featurizer $\Phi_\epsilon$ which, combined with the ERM-optimal classifier $\hat{\beta} = [\beta_c, \beta_{e;\text{ERM}}, \beta_0]^T$, satisfies the following properties, where we define $p_\epsilon := \exp\{-d_e \min(\epsilon - 1, (\epsilon - 1)^2)/8\}$):*

1. *The regularization term of $\Phi_\epsilon, \hat{\beta}$ as in Equation 5 is bounded as*

$$\frac{1}{E} \sum_{e \in \mathcal{E}} \|\nabla_{\hat{\beta}} \mathcal{R}^e(\Phi_\epsilon, \hat{\beta})\|_2^2 \in \mathcal{O}\left( p_\epsilon^2 \left( c_\epsilon d_e + \overline{\|\mu\|_2^2} \right) \right),$$

   *for some constant $c_\epsilon$ that depends only on $\epsilon$.*

2. *$\Phi_\epsilon, \hat{\beta}$ exactly matches the optimal invariant predictor on at least a $1 - p_\epsilon$ fraction of the training set. On the remaining inputs, it matches the ERM-optimal solution.*

*Further, for any test distribution, suppose its environmental mean $\mu_{E+1}$ is sufficiently far from the training means:*

$$\forall e \in \mathcal{E}, \; \min_{y \in \{\pm 1\}} \|\mu_{E+1} - y \cdot \mu_e\|_2 \geq (\sqrt{\epsilon} + \delta)\sqrt{d_e} \tag{7}$$

*for some $\delta > 0$, and define $q := \frac{2E}{\sqrt{\pi}\delta} \exp\{-\delta^2\}$. Then the following holds:*

3. *$\Phi_\epsilon, \hat{\beta}$ is equivalent to the ERM-optimal predictor on at least a $1 - q$ fraction of the test distribution.*

4. *Under Assumption 1, suppose it holds that $\mu_{E+1} = -\sum_{e \in \mathcal{E}} \alpha_e \mu_e$ for some set of coefficients $\{\alpha_e\}_{e \in \mathcal{E}}$. Then so long as*

$$\sum_{e \in \mathcal{E}} \alpha_e \|\mu_e\|_2^2 \geq \frac{\|\mu_c\|_2^2 / \sigma_c^2 + |\beta_0|/2 + \sigma_{\text{ERM}}}{1 - \gamma}, \tag{8}$$

   *the 0-1 risk of $\Phi_\epsilon, \hat{\beta}$ on the new environment is greater than $.975 - q$.*

We give a brief intuition for each of the claims made in this theorem, followed by a sketch of the proof—the full proof can be found in Appendix D.

1. The first claim says that the predictor we construct will have a gradient squared norm scaling as $p_\epsilon^2$ which is exponentially small in $d_e$. Thus, in high dimensions, it will appear as a perfectly reasonable solution to the objective (5).

2. The second claim says that this predictor is identical to the invariant optimal predictor on all but an exponentially small fraction of the training data; on the remaining fraction, it matches the ERM-optimal solution, which has lower risk. The correspondence between constrained and penalized optimization implies that for large enough $d_e$, the "fake" predictor will often be a preferred solution. In the finite-sample setting, we would need exponentially many samples to even distinguish between the two!

3. The third claim is the crux of the theorem; it says that this predictor we've constructed will completely fail to use invariant prediction on most environments. Recall, the intent of IRM is to perform well precisely when ERM breaks down: when the test distribution differs greatly from the training distribution. Assuming a Gaussian prior on the training environment means, they will have separation in $\mathcal{O}(\sqrt{d_e})$ with high probability. Observe that $q$ will be vanishingly small so long as $\delta \geq \text{polylog}(E)$. Part 3 says that IRM fails to use invariant prediction on any environment that is slightly outside the high probability region of the prior; even a separation of $\Omega(\sqrt{d_e \log E})$ suffices. If we expect the new environments to be similar, ERM already guarantees reasonable performance at test-time; thus, IRM fundamentally *does not improve* over ERM in this regime.

4. The final statement demonstrates a particularly egregious failure case of this predictor: just like ERM, if the correlation between the non-invariant features and the label reverses at test-time, our predictor will have significantly worse than chance performance.

*Proof Sketch.* We give a construction which is almost identical to the optimal invariant predictor on the training data yet behaves like the ERM solution at test time. We partition the environmental feature space into two sets, $\mathcal{B}, \mathcal{B}^c$. $\mathcal{B}$ is the union of balls centered at each $\mu_e$, each with a large enough radius that it contains most of the samples from that environment; thus $\mathcal{B}$ represents the vast majority of the training distribution. On this set, define $\Phi(x) = [z_c]$, so our construction is equal to the optimal invariant predictor. Now consider $\mathcal{B}^c = \mathbb{R}^{d_e} \setminus \mathcal{B}$. We use standard concentration results to upper bound the measure of $\mathcal{B}^c$ under the training distribution by $p$. Next, we show how choosing $\Phi(x) = f^{-1}(x) = [z_c, z_e]^T$ on this set results in the sub-optimality bound, which is of order $p^2$. It is also clear that our constructed predictor is equivalent to the ERM-optimal solution on $\mathcal{B}^c$. Thus, our predictor will often have lower empirical risk on $\mathcal{B}^c$, countering the regularization penalty.

The second part of the proof shows that while $\mathcal{B}$ has large measure under the training environments, it will have very small measure under any moderately different test environment. We can see this by considering the separation of means (Equation 7); the measure of each ball in $\mathcal{B}$ can be bounded by the measure of the halfspace containing it; if each ball is far enough away from $\mu_{E+1}$, then the total measure of these halfspaces must be small. At test time, our predictor will therefore match the ERM solution on all but $q$ of the observations (part 3). Finally, we lower bound the 0-1 risk of the ERM predictor under such a distribution shift by analyzing the distribution of the response variable. The proof is completed by observing that our predictor's risk can differ from this by at most $q$. $\square$

Theorem 6.1 shows that it's possible for the IRM solution to perform poorly on environments which differ even moderately from the training data. We can of course guarantee generalization if the training distributions "cover" (or approximately cover) the full space of environments in order to tie down the performance on future distributions. But in such a scenario, there would no longer be a need for ICP; we could expect ERM or DRO to perform just as well. Once more, we find that our result trivially extends to the alternative objectives; we again refer to Appendix E.

## 7 CONCLUSION

Out-of-distribution generalization is an important direction for future research, and Invariant Causal Prediction remains a promising approach. However, formal results for latent variable models are lacking, particularly in the non-linear setting with fully unobserved covariates. This paper demonstrates that Invariant Risk Minimization and subsequent related works have significant under-explored risks and issues with their formulation. This raises the question: what is the correct formulation for invariant prediction when the observations are complex, non-linear functions of unobserved latent factors? We hope that this work will inspire further theoretical study on the effectiveness of IRM and similar objectives for invariant prediction.

ACKNOWLEDGEMENTS

We thank Adarsh Prasad, Jeremy Cohen, and Zack Lipton for helpful feedback. Special thanks to Adarsh Prasad for noticing our initial formatting error. E.R. and P.R. acknowledge the support of NSF via IIS-1909816, IIS-1955532 and ONR via N000141812861.

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

## A    ADDITIONAL NOTATION FOR THE APPENDIX

To avoid overloading, we use $\phi, F$ for the standard Gaussian PDF and CDF respectively. We write $\mathcal{S}^c$ to denote the set complement of a set $\mathcal{S}$. We write $|\cdot|$ to denote entrywise absolute value.

## B    PROOF OF PROPOSITION 4.1

Recall the IRM objective:

$$\min_{\Phi, \beta} \quad \mathbb{E}_{(x,y)\sim p(x,y)}[-\log \sigma(y \cdot \hat{\beta}^T \Phi(x))]$$

$$\text{subject to} \quad \frac{\partial}{\partial \hat{\beta}} \mathbb{E}_{(x,y)\sim p^e}[-\log \sigma(y \cdot \hat{\beta}^T \Phi(x))] = 0. \ \forall e \in \mathcal{E}.$$

Concretely, we represent $\Phi$ as some parametrized function $\Phi_\theta$, over whose parameters $\theta$ we then optimize. The derivative of the negative log-likelihood for logistic regression with respect to the $\beta$ coefficients is well known:

$$\frac{\partial}{\partial \hat{\beta}}\left[-\log \sigma(y \cdot \hat{\beta}^T \Phi_\theta(x))\right] = (\sigma(\hat{\beta}^T \Phi_\theta(x)) - \mathbf{1}\{y = 1\})\Phi_\theta(x).$$

Suppose we recover the true invariant features $\Phi_\theta(x) = \begin{bmatrix} z_c \\ \mathbf{0} \end{bmatrix}$ and coefficients $\hat{\beta} = \begin{bmatrix} \beta \\ \mathbf{0} \end{bmatrix}$ (in other words, we allow for the introduction of new features). Then the IRM constraint becomes:

$$0 = \frac{\partial}{\partial \hat{\beta}} \mathbb{E}_{(x,y)\sim p^e}[-\log \sigma(y \cdot \hat{\beta}^T \Phi_\theta(x))]$$

$$= \int_{\mathcal{Z}} p^e(z_c) \sum_{y \in \{\pm 1\}} p^e(y \mid z_c) \frac{\partial}{\partial \hat{\beta}} \left[-\log \sigma(y \cdot \beta^T z_c)\right] \ dz_c$$

$$= \int_{\mathcal{Z}} p^e(z_c)\Phi_\theta(x)\left[\sigma(\hat{\beta}^T z_c)(\sigma(\hat{\beta}^T z_c) - 1) + (1 - \sigma(\hat{\beta}^T z_c))\sigma(\hat{\beta}^T z_c)\right] dz_c.$$

Since $\hat{\beta}$ is constant across environments, this constraint is clearly satisfied for every environment, and is therefore also the minimizing $\hat{\beta}$ for the training data as a whole.

Considering now the derivative with respect to the featurizer $\Phi_\theta$:

$$\frac{\partial}{\partial \theta}\left[-\log \sigma(y \cdot \hat{\beta}^T \Phi_\theta(x))\right] = (\sigma(\hat{\beta}^T \Phi_\theta(x)) - \mathbf{1}\{y = 1\})\frac{\partial}{\partial \theta}\hat{\beta}^T \Phi_\theta(x).$$

Then the derivative of the loss with respect to these parameters is

$$\int_{\mathcal{Z}} p^e(z_c)\left(\frac{\partial}{\partial \theta}\hat{\beta}^T \Phi_\theta(x)\right)\left[\sigma(\beta^T z_c)(\sigma(\beta^T z_c) - 1) + (1 - \sigma(\beta^T z_c))\sigma(\beta^T z_c)\right] dz_c = 0.$$

So, the optimal invariant predictor is a stationary point with respect to the feature map parameters as well.

## C    RESULTS FROM SECTION 5

### C.1    PROOF OF THEOREM 5.1

We begin by formally stating the non-degeneracy condition. Consider any environmental mean $\mu_e$, and suppose it can be written as a linear combination of the others means with coefficients $\alpha^e$:

$$\mu_e = \sum_i \alpha_i^e \mu_i.$$

Then the environments are considered non-degenerate if the following inequality holds for any such set of coefficients:

$$\sum_i \alpha_i^e \neq 1, \tag{9}$$

and furthermore that the following ratio is different for at least two different environments $a, b$:

$$\exists \alpha^a, \alpha^b. \frac{\sigma_a^2 - \sum_i \alpha_i^a \sigma_i^2}{1 - \sum_i \alpha_i^a} \neq \frac{\sigma_b^2 - \sum_i \alpha_i^b \sigma_i^2}{1 - \sum_i \alpha_i^b}. \tag{10}$$

The first inequality says that none of the environmental means are are an affine combination of the others; in other words, they lie in *general linear position*, which is the same requirement as Arjovsky et al. (2019). The other inequality is a similarly lax non-degeneracy requirement regarding the relative scale of the variances. It is clear that the set of environmental parameters that do not satisfy Equations 9 and 10 has measure zero under any absolutely continuous density, and similarly, if $E \leq d_e$ then the environmental means will be linearly independent almost surely.

We can now proceed with the proof, beginning with some helper lemmas:

**Lemma C.1.** *Suppose we observe $E$ environments $\mathcal{E} = \{e_1, e_2, \ldots, e_E\}$, each with environmental mean of dimension $d_e \geq E$, such that all environmental means are linearly independent. Then there is a unique unit-norm vector $p$ such that*

$$p^T \mu_e = \sigma_e^2 \tilde{\mu} \ \forall e \in \mathcal{E}, \tag{11}$$

*where $\tilde{\mu}$ is the largest scalar which admits such a solution.*

*Proof.* Let $v_1, v_2, \ldots, v_E$ be a set of basis vectors for $\text{span}\{\mu_1, \mu_2, \ldots, \mu_E\}$. Each mean can then be expressed as a combination of these basis vectors: $u_i = \sum_{j=1}^E \alpha_{ij} v_j$. Since the means are linearly independent, we can combine these coefficients into a single invertible matrix

$$U = \begin{bmatrix} \alpha_{11} & \alpha_{21} & \ldots & \alpha_{E1} \\ \alpha_{12} & \alpha_{22} & \ldots & \alpha_{E2} \\ \vdots & \vdots & \ddots & \vdots \\ \alpha_{1E} & \alpha_{2E} & \ldots & \alpha_{EE} \end{bmatrix}.$$

We can then combine the constraints (11) as

$$U^T p_\alpha = \boldsymbol{\sigma} \triangleq \begin{bmatrix} \sigma_1^2 \\ \sigma_2^2 \\ \vdots \\ \sigma_E^2 \end{bmatrix},$$

where $p_\alpha$ denotes our solution expressed in terms of the basis vectors $\{v_i\}_{i=1}^E$. This then has the solution

$$p_\alpha = U^{-T} \boldsymbol{\sigma}.$$

This defines the entire space of solutions, which consists of $p_\alpha$ plus any element of the remaining $(d_e - E)$-dimensional orthogonal subspace. However, we want $p$ to be unit-norm—observe that the current vector solves Equation 11 with $\tilde{\mu} = 1$, which means that after normalizing we get $\tilde{\mu} = \frac{1}{\|p_\alpha\|_2}$. Adding any element of the orthogonal subspace would only increase the norm of $p$, decreasing $\tilde{\mu}$. Thus, the unique maximizing solution is

$$p_\alpha = \frac{U^{-T} \boldsymbol{\sigma}}{\|U^{-T} \boldsymbol{\sigma}\|_2}, \quad \text{with} \quad \tilde{\mu} = \frac{1}{\|U^{-T} \boldsymbol{\sigma}\|_2}.$$

Finally, $p_\alpha$ has to be rotated back into the original space by defining $p = \sum_{i=1}^E p_{\alpha i} v_i$. $\qquad \square$

**Lemma C.2.** *Assume $f$ is linear. Suppose we observe $E \leq d_e$ environments whose means are linearly independent. Then there exists a linear $\Phi$ with $\text{rank}(\Phi) = d_c + d_e + 1 - E$ whose output depends on the environmental features, yet the optimal classifier on top of $\Phi$ is invariant.*

*Proof.* We will begin with the case when $E = d_e$ and then show how to modify this construction for when $E < d_e$. Consider defining

$$\Phi = \begin{bmatrix} I & 0 \\ 0 & M \end{bmatrix} \circ f^{-1}$$

with

$$M = \begin{bmatrix} — & p^T & — \\ — & 0 & — \\ & \vdots & \\ — & 0 & — \end{bmatrix}.$$

Here, $p \in \mathbb{R}^{d_c}$ is defined as the unit-norm vector solution to

$$p^T \mu_e = \sigma_e^2 \tilde{\mu} \quad \forall e$$

such that $\tilde{\mu}$ is maximized—such a vector is guaranteed to exist by Lemma C.1. Thus we get $\Phi(x) = \begin{bmatrix} z_c \\ p^T z_e \end{bmatrix}$, which is of rank $d_c + 1$ as desired. Define $\tilde{z}_e = p^T z_e$, which means that $\tilde{z}_e \mid y \sim \mathcal{N}(y \cdot \sigma_e^2 \tilde{\mu}, \sigma_e^2)$. For each environment we have

$$
\begin{aligned}
p(y \mid z_c, \tilde{z}_e) &= \frac{p(z_c, \tilde{z}_e, y)}{p(z_c, \tilde{z}_e)} \\
&= \frac{\sigma(y \cdot \beta_c^T z_c) p(\tilde{z}_e \mid y \cdot \sigma_e^{e2} \tilde{\mu}, \sigma_e^2)}{[\sigma(y \cdot \beta_c^T z_c) p(\tilde{z}_e \mid y \cdot \sigma_e^{e2} \tilde{\mu}, \sigma_e^2) + \sigma(-y \cdot \beta_c^T z_c) p(\tilde{z}_e \mid -y \cdot \sigma_e^{e2} \tilde{\mu}, \sigma_e^2)]} \\
&= \frac{\sigma(y \cdot \beta_c^T z_c) \exp(y \cdot \tilde{z}_e \tilde{\mu})}{[\sigma(y \cdot \beta_c^T z_c) \exp(y \cdot \tilde{z}_e \tilde{\mu}) + \sigma(-y \cdot \beta_c^T z_c) \exp(-y \cdot \tilde{z}_e \tilde{\mu})]} \\
&= \frac{1}{1 + \exp(-y \cdot (\beta_c^T z_c + 2 \tilde{z}_e \tilde{\mu}))}.
\end{aligned}
$$

The log-odds of $y$ is linear in these features, so the optimal classifier is

$$\hat{\beta} = \begin{bmatrix} \beta_c \\ 2\tilde{\mu} \end{bmatrix},$$

which is the same for all environments.

Now we show how to modify this construction for when $E < d_e$. If we remove one of the environmental means, $\Phi$ trivially maintains its feasibility. Note that since they are linearly independent, the mean which was removed has a component in a direction orthogonal to the remaining means. Call this component $p'$, and consider redefining $M$ as

$$M = \begin{bmatrix} — & p^T & — \\ — & p'^T & — \\ — & 0 & — \\ & \vdots & \\ — & 0 & — \end{bmatrix}.$$

The distribution of $\tilde{z}_e$ in each of the remaining dimensions is normal with mean 0, which means a corresponding coefficient of 0 is optimal for all environments. So the classifier $\hat{\beta}$ remains optimal for all environments, yet we've added another row to $M$ which increases the dimensionality of its span, and therefore the rank of $\Phi$, by 1. Working backwards, we can repeat this process for each additional mean, such that $\text{rank}(\Phi) = d_c + 1 + (d_e - E)$, as desired. Note that for $E = 1$ any $\Phi$ will be vacuously feasible. $\square$

**Lemma C.3.** *Suppose we observe $E$ environments $\mathcal{E} = \{e_1, e_2, \ldots, e_E\}$ whose parameters satisfy the non-degeneracy conditions (9, 10). Let $\Phi(x) = Az_c + Bz_e$ be any feature vector which is a linear function of the invariant and environmental features, and suppose the optimal $\hat{\beta}$ on top of $\Phi$ is invariant. If $E > d_e$, then $\hat{\beta} = 0$ or $B = 0$.*

*Proof.* Write $\Phi = [A|B]$ where $A \in \mathbb{R}^{d \times d_c}, B \in \mathbb{R}^{d \times d_e}$ and define

$$\bar{\mu}_e = \Phi \begin{bmatrix} \mu_c \\ \mu_e \end{bmatrix} \qquad\qquad = A\mu_c + B\mu_e,$$

$$\bar{\Sigma}_e = \Phi \begin{bmatrix} \sigma_c^2 I_{d_c} & 0 \\ 0 & \sigma_e^2 I_{d_e} \end{bmatrix} \Phi^T \qquad = \sigma_c^2 A A^T + \sigma_e^2 B B^T.$$

Without loss of generality we assume $\bar{\Sigma}$ is invertible (if it is not, we can consider just the subspace in which it is—outside of this space, the features have no variance and therefore cannot carry information about the label). By Lemma F.2, the optimal coefficient for each environment is $2\bar{\Sigma}_e^{-1}\bar{\mu}_e$. In order for this vector to be invariant, it must be the same across environments; we write it as a constant $\hat{\beta}$. Suppose $\bar{\mu}_e = 0$ for some environment $e$—then the claim is trivially true with $\hat{\beta} = 0$. We therefore proceed under the assumption that $\bar{\mu}_e \neq 0 \; \forall e \in \mathcal{E}$.

With this fact, we have that $\forall e \in \mathcal{E}$,

$$\hat{\beta} = 2(\sigma_c^2 A A^T + \sigma_e^2 B B^T)^{-1}(A\mu_c + B\mu_e)$$
$$\iff (\sigma_c^2 A A^T + \sigma_e^2 B B^T)\hat{\beta} = 2A\mu_c + 2B\mu_e$$
$$\iff \sigma_e^2 B B^T \hat{\beta} - 2B\mu_e = 2A\mu_c - \sigma_c^2 A A^T \hat{\beta}. \tag{12}$$

Define the vector $\mathbf{v} = 2A\mu_c - \sigma_c^2 A A^T \hat{\beta}$. We will show that for any $\hat{\beta}, A$, with probability 1 only $B = 0$ can satisfy Equation 12 for every environment. If $E > d_e$, then there exists at least one environmental mean which can be written as a linear combination of the others. Without loss of generality, denote the parameters of this environment as $(\bar{\mu}, \bar{\sigma}^2)$ and write $\bar{\mu} = \sum_{i=1}^{d_e} \alpha_i \mu_i$. However, note that by assumption the means lie in general linear position, and therefore we actually have at least $d_e$ sets of coefficients $\alpha$ for which this holds. Rearranging Equation 12, we get

$$\bar{\sigma}^2 B B^T \hat{\beta} - \mathbf{v} = 2B\bar{\mu}$$

$$= \sum_{i=1}^{d_e} \alpha_i 2B\mu_i$$

$$= \sum_{i=1}^{d_e} \alpha_i \left[ \sigma_i^2 B B^T \hat{\beta} - \mathbf{v} \right],$$

and rearranging once more yields

$$\left( \bar{\sigma}^2 - \sum \alpha_i \sigma_i^2 \right) B B^T \hat{\beta} = \left( 1 - \sum \alpha_i \right) \mathbf{v}.$$

By assumption, $(1 - \sum \alpha_i)$ is non-zero. We can therefore rewrite this as

$$\hat{\alpha} B B^T \hat{\beta} = \mathbf{v},$$

where $\hat{\alpha} = \frac{\bar{\sigma}^2 - \sum \alpha_i \sigma_i^2}{1 - \sum \alpha_i}$ is a scalar. As the vectors $B B^T \hat{\beta}$ and $\mathbf{v}$ are both independent of the environment, this can only hold true if $\hat{\alpha}$ is fixed for all environments or if both $B B^T \hat{\beta}, \mathbf{v}$ are 0. The former is false by assumption, so the the latter must hold.

As a result, we see that Equation 12 reduces to

$$B\mu_e = 0 \quad \forall e \in \mathcal{E}.$$

As the span of the observed $\mu_e$ is all of $\mathbb{R}^{d_e}$, this is only possible if $B = 0$. $\qquad\square$

We are now ready to prove the main claim. We restate the theorem here for convenience:

**Theorem 5.1** (Linear case)**.** *Assume $f$ is linear. Suppose we observe $E$ training environments. Then the following hold:*

1. *Suppose $E > d_e$. Consider any linear featurizer $\Phi$ which is feasible under the IRM objective (4), with invariant optimal classifier $\hat{\beta} \neq 0$, and write $\Phi(f(z_c, z_e)) = Az_c + Bz_e$. Then under mild non-degeneracy conditions, it holds that $B = 0$. Consequently, $\hat{\beta}$ is the optimal classifier for all possible environments.*

2. *If $E \leq d_e$ and the environmental means $\mu_e$ are linearly independent, then there exists a linear $\Phi$—where $\Phi(f(z_c, z_e)) = Az_c + Bz_e$ with $rank(B) = d_e + 1 - E$—which is feasible under the IRM objective. Further, both the logistic and 0-1 risks of this $\Phi$ and its corresponding optimal $\hat{\beta}$ are strictly lower than those of the optimal invariant predictor.*

*Proof.*      1. Since $\Phi, f$ are linear, we can write $\Phi(x) = Az_c + Bz_e$ for some matrices $A, B$. Assume the non-degeneracy conditions (9, 10) hold. By Lemma C.3, one of $B = 0$ or $\hat{\beta} = 0$ holds. Thus, $\Phi, \hat{\beta}$ uses only invariant features. Since the joint distribution $p^e(z_c, y)$ is invariant, this predictor has identical risk across all environments.

2. The existence of such a predictor is proven by Lemma C.2. It remains to show that the risk of this discriminator is lower than that of the optimal invariant predictor. Observe that these features are non-degenerate independent random variables with support over all of $\mathbb{R}$, and therefore by Lemma F.1, dropping the $\tilde{z}_e$ term and using

$$\Phi(x) = [z_c], \quad \hat{\beta} = \begin{bmatrix} \beta_c \\ \beta_0 \end{bmatrix}$$

results in strictly higher risk. The proof is completed by noting that this definition is precisely the optimal invariant predictor. $\qquad\square$

## C.2    Experiments for Theorem 5.1

To corroborate our theoretical findings, we run an experiment on data drawn from our model to see at what point IRM is able to recover a generalizing predictor. We generated data precisely according to our model in the linear setting, with $d_c = 3, d_e = 6$. The environmental means were drawn from a multivariate Gaussian prior; we randomly generated the invariant parameters and the parameters of the prior such that using the invariant features gave reasonable accuracy (71.9%) but the environmental features would allow for almost perfect accuracy on in-distribution test data (99.8%). Thus, the goal was to see if IRM could successfully learn a predictor which ignores meaningful covariates $z_e$, to the detriment of its training performance but to the benefit of OOD generalization. We chose equal class marginals ($\eta = 0.5$).

Figure C.1 shows the result of five runs of IRM, each with different environmental parameters but the same invariant parameters (the training data itself was redrawn for each run). We found that optimizing for the IRM objective was quite unstable, frequently collapsing to the ERM solution unless $\lambda$ and the optimizer learning rate were carefully tuned. This echoes the results of Krueger et al. (2020) who found that tuning $\lambda$ during training to specific values at precisely the right time is essential for good performance. To prevent collapse, we kept the same environmental prior and found a single setting for $\lambda$ and the learning rate which resulted in reasonable performance across all five runs. At test time, we evaluated the trained predictors on additional, unseen environments whose parameters were drawn from the same prior. To simulate distribution shift, we evaluated the predictors on the same data but with the environmental means negated. Thus the correlations between the environmental features $z_e$ and the label $y$ were reversed.

Observe that the results closely track the expected outcome according to Theorem 5.1: up until $E = d_e$, IRM essentially matches ERM in performance both in-distribution and under distribution shift. As soon as we cross that threshold of observed environments, the predictor learned via IRM begins to perform drastically better under distribution shift, behaving more like the optimal invariant predictor. We did however observe that occasionally the invariant solution would be found after only $E = d_e = 6$ environments; we conjecture that this is because at this point the feasible-yet-not-invariant predictor with lower objective value presented in Theorem 5.1 is precisely a single point, as opposed to a multi-dimensional subspace, and therefore might be difficult for the optimizer to find.

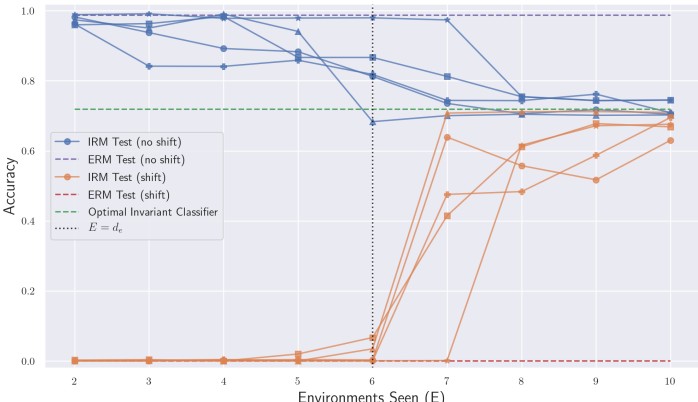

Figure C.1: Performance of predictors learned with IRM (5 different runs) and ERM (dashed lines) on test distributions where the correlation between environmental features and the label is consistent (no shift) or reversed (shift). The dashed green line is the performance of the optimal invariant predictor. Observe that up until $E = d_e$, IRM consistently returns a predictor with performance similar to ERM: good generalization without distribution shift, but catastrophic failure when the correlation is reversed. In contrast, once $E > d_e$, IRM is able to recover a $\Phi, \hat{\beta}$ with performance similar to that of the invariant optimal predictor.

## C.3 PROOF OF THEOREM 5.3

**Theorem 5.3.** *Suppose we observe $E \leq d_e$ environments, such that all environmental means are linearly independent. Then there exists a feasible $\Phi, \hat{\beta}$ which uses* only *environmental features and achieves lower 0-1 risk than the optimal invariant predictor on every environment $e$ such that $\sigma_e \tilde{\mu} > \sigma_c^{-1} \|\mu_c\|_2$ and $2\sigma_e \tilde{\mu} \sigma_c^{-1} \|\mu_c\|_2 \geq |\beta_0|$.*

*Proof.* We consider the non-invariant predictor constructed as described in Lemma C.2, but dropping the invariant features and coefficients. By Lemma F.2, the optimal coefficients for the invariant and non-invariant predictors are

$$\hat{\beta}_{caus} = \begin{bmatrix} 2\sigma_c^{-2}\mu_c \\ \beta_0 \end{bmatrix} \quad \text{and} \quad \hat{\beta}_{non-caus} = \begin{bmatrix} 2\tilde{\mu} \\ \beta_0 \end{bmatrix},$$

respectively. Therefore, the 0-1 risk of the optimal invariant predictor is precisely

$$\eta\mathbb{P}(2\sigma_c^{-2}\mu_c^T z_c + \beta_0 < 0) + (1-\eta)\mathbb{P}(-2\sigma_c^{-2}\mu_c^T z_c + \beta_0 > 0)$$
$$=\eta F\left(-\sigma_c^{-1}\|\mu_c\|_2 - \frac{\beta_0 \sigma_c}{2\|\mu_c\|_2}\right) + (1-\eta)F\left(-\sigma_c^{-1}\|\mu_c\|_2 + \frac{\beta_0 \sigma_c}{2\|\mu_c\|_2}\right),$$

where $F$ is the Gaussian CDF. By the same reasoning, the 0-1 risk of the non-invariant predictor is

$$\eta F\left(-\sigma_e \tilde{\mu} - \frac{\beta_0}{2\sigma_e \tilde{\mu}}\right) + (1-\eta)F\left(-\sigma_e \tilde{\mu} + \frac{\beta_0}{2\sigma_e \tilde{\mu}}\right).$$

Define $\alpha = \sigma_c^{-1}\|\mu_c\|_2$ and $\gamma = \sigma_e \tilde{\mu}$. By monotonicity of the Gaussian CDF, the former risk is higher than the latter if

$$\alpha + \frac{\beta_0}{2\alpha} \leq \gamma + \frac{\beta_0}{2\gamma}, \tag{13}$$

$$\alpha - \frac{\beta_0}{2\alpha} < \gamma - \frac{\beta_0}{2\gamma}. \tag{14}$$

Without loss of generality, we will prove these inequalities for $\beta_0 \geq 0$; an identical argument proves it for $\beta_0 < 0$ but with the '$\leq$' and '$<$' swapped.

Suppose $\gamma > \alpha$ (the first condition). Then Equation 14 is immediate. Finally, for Equation 13, observe that

$$\gamma + \frac{\beta_0}{2\gamma} \geq \alpha + \frac{\beta_0}{2\alpha}$$

$$\iff \gamma - \alpha \geq \frac{\beta_0}{2\alpha} - \frac{\beta_0}{2\gamma} = \frac{(\gamma - \alpha)\beta_0}{2\gamma\alpha}$$

$$\iff 2\gamma\alpha \geq \beta_0,$$

which is the second condition. $\qquad\square$

### C.4 SIMULATIONS OF MAGNITUDE OF ENVIRONMENTAL FEATURES

As discussed in Section 5, analytically quantifying the solution $\tilde{\mu}$ to the equation in Lemma C.1 is difficult; instead, we present simulations to give a sense of how often these conditions would hold in practice.

For each choice of environmental dimension $d_e$, we generated a "base" correlation $b \sim \mathcal{N}(0, I_{d_e})$ as the mean of the prior over environmental means $\mu_e$. Each of these $\mu_e$ was then drawn from $\mathcal{N}(b, 4I_{d_e})$—thus, while they all came from the same prior, the noise in the draw of each $\mu_e$ was significantly larger than the bias induced by the prior. We then solved for the precise value $\sigma_e\tilde{\mu}$, with the same variance $\sigma_e^2$ for all environments, chosen as a hyperparameter. The shaded area represents a 95% confidence interval over 20 runs.

The dotted lines are $\sqrt{d_c}$. If we imagine the invariant parameters are drawn from a standard Gaussian prior, then this is precisely $\mathbb{E}[\sigma_c^{-1}\|\mu_c\|_2]$. Thus, the point where $\sigma_e\tilde{\mu}$ crosses these dotted lines is approximately how many environments would need to be observed before the non-invariant predictor has higher risk than the optimal invariant predictor. We note that this value is quite large, on the order of $d_e - d_c$.

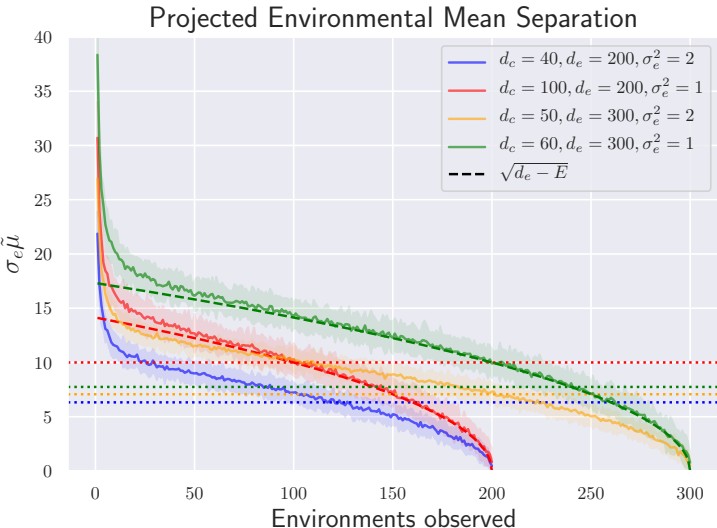

Figure C.2: Simulations to evaluate $\sigma_e\tilde{\mu}$ for varying ratios of $\frac{d_e}{d_c}$. When $\sigma_e^2 = 1$ the value closely tracks $\sqrt{d_e - E}$, and the crossover point is approximately $d_e - \sigma_e^2 d_c$. These results imply the conditions of Theorem 5.3 are very likely to hold in the high-dimensional setting.

## D THEOREM 6.1 AND DISCUSSION

### D.1 PROOF OF THEOREM 6.1

We again begin with helper lemmas.

Our featurizer $\Phi$ is constructed to recover the environmental features only if they fall within a set $\mathcal{B}^c$. The following lemma shows that since only the environmental features contribute to the gradient penalty, the penalty can be bounded as a function of the measure and geometry of that set. This is used together with Lemmas F.3 and F.4 to bound the overall penalty of our constructed predictor.

**Lemma D.1.** *Suppose we observe environments $\mathcal{E} = \{e_1, e_2, \ldots\}$. Given a set $\mathcal{B} \subseteq \mathbb{R}^{d_e}$, consider the predictor defined by Equation 19. Then for any environment $e$, the penalty term of this predictor in Equation 5 is bounded as*

$$\|\nabla_{\hat{\beta}} \mathcal{R}^e(\Phi, \hat{\beta})\|_2^2 \leq \left\| \mathbb{P}(z_e \in \mathcal{B}^c) \mathbb{E}[|z_e| \mid z_e \in \mathcal{B}^c] \right\|_2^2.$$

*Proof.* We write out the precise form of the gradient for an environment $e$:

$$\nabla_{\hat{\beta}} \mathcal{R}^e(\Phi, \hat{\beta}) = \int_{\mathcal{Z}_c \times \mathcal{Z}_e} p^e(z_c, z_e) \left[ \sigma(\hat{\beta}^T \Phi(f(z_c, z_e))) - p^e(y = 1 \mid z_c, z_e) \right] \Phi(f(z_c, z_e)) \, d(z_c, z_e).$$

Observe that since $z_c \perp\!\!\!\perp z_e \mid y$, the optimal invariant coefficients are unchanged, and therefore the gradient in the invariant dimensions is 0. We can split the gradient in the environmental dimensions into two integrals:

$$\int_{\mathcal{Z}_c \times \mathcal{B}} p^e(z_c, z_e) \left[ \sigma(\beta_c^T z_c + \beta_0) - p^e(y = 1 \mid z_c, z_e) \right] [0] \, d(z_c, z_e)$$

$$+ \int_{\mathcal{Z}_c \times \mathcal{B}^c} p^e(z_c, z_e) \left[ \sigma(\beta_c^T z_c + \beta_{e;\text{ERM}}^T z_e + \beta_0) - \sigma(\beta_c^T z_c + \beta_e^T z_e + \beta_0) \right] [z_e] \, d(z_c, z_e).$$

Since the features are 0 within $\mathcal{B}$, the first term reduces to 0. For the second term, note that $\forall\, x, y \in \mathbb{R}$, $|\sigma(x) - \sigma(y)| \leq 1$, and therefore

$$|\nabla_{\hat{\beta}} \mathcal{R}^e(\Phi, \hat{\beta})| \leq \int_{\mathcal{Z}_c \times \mathcal{B}^c} p^e(z_c, z_e)[|z_e|] \, d(z_c, z_e).$$

We can marginalize out $z_c$, and noting that we want to bound the squared norm,

$$\|\nabla_{\hat{\beta}} \mathcal{R}^e(\Phi, \hat{\beta})\|_2^2 \leq \left\| \int_{\mathcal{B}^c} p^e(z_e)[|z_e|] \, dz_e \right\|_2^2$$

$$= \left\| \mathbb{P}(z_e \in \mathcal{B}^c) \mathbb{E}[|z_e| \mid z_e \in \mathcal{B}^c] \right\|_2^2. \qquad \square$$

This next lemma says that if the environmental mean of the test distribution is sufficiently separated from each of the training means, with high probability a sample from this distribution will fall outside of $\mathcal{B}_r$, and therefore $\Phi_\epsilon, \hat{\beta}$ will be equivalent to the ERM solution.

**Lemma D.2.** *For a set of $E$ environments $\mathcal{E} = \{e_1, e_2, \ldots, e_E\}$ and any $\epsilon > 1$, construct $\mathcal{B}_r$ as in Equation 18 and define $\Phi_\epsilon$ using $\mathcal{B}_r$ as in Equation 19. Suppose we now test on a new environment with parameters $(\mu_{E+1}, \sigma_{E+1}^2)$, and assume Equation 15 holds with parameter $\delta$. Define $k = \min_{e \in \mathcal{E}} \frac{\sigma_e^2}{\sigma_{E+1}^2}$. Then with probability $\geq 1 - \frac{2E}{\sqrt{k\pi}\delta} \exp\{-k\delta^2\}$ over the draw of an observation from this new environment, we have*

$$\Phi_\epsilon(x) = f^{-1}(x) = \begin{bmatrix} z_c \\ z_e \end{bmatrix}.$$

*Proof.* By Equation 15 our new environmental mean is sufficiently far away from all the label-conditional means of the training environments. In particular, for any environment $e \in \mathcal{E}$ and any label $y \in \{\pm 1\}$, the $\ell_2$ distance from that mean to $\mu_{E+1}$ is at least $(\sqrt{\epsilon} + \delta)\sigma_e \sqrt{d_e}$.

Recall that $\mathcal{B}_r$ is the union of balls $\pm B_e$, where $B_e$ is the ball of $\ell_2$ radius $\sqrt{\epsilon \sigma_e^2 d_e}$ centered at $\mu_e$. For each environment $e$, consider constructing the halfspace which is perpendicular to the line connecting

$\mu_e$ and $\mu_{E+1}$ and tangent to $B_e$. This halfspace fully contains $B_e$, and therefore the measure of $B_e$ is upper bounded by that of the halfspace.

By rotational invariance of the Gaussian distribution, we can rotate this halfspace into one dimension and the measure will not change. The center of the ball is $(\sqrt{\epsilon} + \delta)\sigma_e\sqrt{d_e}$ away from the mean $\mu_{E+1}$, so accounting for its radius, the distance from the mean to the halfspace is $\delta\sigma_e\sqrt{d_e}$. The variance of the rotated distribution one dimension is $\sigma_{E+1}^2 d_e$, so the measure of this halfspace is upper bounded by

$$1 - \Phi\left(\frac{\delta\sigma_e\sqrt{d_e}}{\sqrt{\sigma_{E+1}^2 d_e}}\right) \leq \Phi\left(-\sqrt{k}\delta\right)$$

$$\leq \frac{1}{\sqrt{k\pi}\delta}\exp\{-k\delta^2\},$$

using results from Kschischang (2017). There are $2E$ such balls comprising $\mathcal{B}_r$, which can be combined via union bound. $\qquad\square$

With these two lemmas, we now state the full version of Theorem 6.1, with the main difference being that it allows for any environmental variance.

**Theorem D.3** (Non-linear case, full). *Suppose we observe $E$ environments $\mathcal{E} = \{e_1, e_2, \ldots, e_E\}$. Then, for any $\epsilon > 1$, there exists a featurizer $\Phi_\epsilon$ which, combined with the ERM-optimal classifier $\hat{\beta} = [\beta_c, \beta_{e;ERM}, \beta_0]^T$, satisfies the following properties, where we define $p_{\epsilon, d_e} := \exp\{-d_e \min((\epsilon - 1), (\epsilon - 1)^2)/8\}$:*

1. *Define $\sigma_{\max}^2 = \max_e \sigma_e^2$. Then the regularization term of $\Phi_\epsilon, \hat{\beta}$ is bounded as*

$$\frac{1}{E}\sum_{e\in\mathcal{E}}\|\nabla_{\hat{\beta}}\mathcal{R}^e(\Phi_\epsilon, \hat{\beta})\|_2^2 \in \mathcal{O}\left(p_{\epsilon, d_e}^2\left[\epsilon d_e\sigma_{\max}^4\exp\{2\epsilon\sigma_{\max}^2\} + \overline{\|\mu\|_2^2}\right]\right).$$

2. *$\Phi_\epsilon, \hat{\beta}$ exactly matches the optimal invariant predictor on at least a $1 - p_{\epsilon, d_e}$ fraction of the training set. On the remaining inputs, it matches the ERM-optimal solution.*

*Further, for any test distribution with environmental parameters $(\mu_{E+1}, \sigma_{E+1}^2)$, suppose the environmental mean $\mu_{E+1}$ is sufficiently far from the training means:*

$$\forall e \in \mathcal{E}, \quad \min_{y\in\{\pm 1\}}\|\mu_{E+1} - y\cdot\mu_e\|_2 \geq (\sqrt{\epsilon} + \delta)\sigma_e\sqrt{d_e} \tag{15}$$

*for some $\delta > 0$. Define the constants:*

$$k = \min_{e\in\mathcal{E}}\frac{\sigma_e^2}{\sigma_{E+1}^2}$$

$$q = \frac{2E}{\sqrt{k\pi}\delta}\exp\{-k\delta^2\}.$$

*Then the following holds:*

3. *$\Phi_\epsilon, \hat{\beta}$ is equivalent to the ERM-optimal predictor on at least a $1 - q$ fraction of the test distribution.*

4. *Under Assumption 1, suppose it holds that*

$$\mu_{E+1} = -\sum_{e\in\mathcal{E}}\alpha_e\mu_e \tag{16}$$

*for some set of coefficients $\{\alpha_e\}_{e\in\mathcal{E}}$. Then for any $c \in \mathbb{R}$, so long as*

$$\sum_{e\in\mathcal{E}}\alpha_e\frac{\|\mu_e\|_2^2}{\sigma_e^2} \geq \frac{\|\mu_c\|_2^2/\sigma_c^2 + |\beta_0|/2 + c\sigma_{ERM}}{1 - \gamma}, \tag{17}$$

*the 0-1 risk of $\Phi_\epsilon, \hat{\beta}$ is lower bounded by $F(2c) - q$.*

*Proof.* Define $r = \sqrt{\epsilon \sigma_e^2 d_e}$ and construct $\mathcal{B}_r \subset \mathbb{R}^{d_e}$ as

$$\mathcal{B}_r = \left[ \bigcup_{e \in \mathcal{E}} B_r(\mu_e) \right] \cup \left[ \bigcup_{e \in \mathcal{E}} B_r(-\mu_e) \right], \tag{18}$$

where $B_r(\alpha)$ is the ball of $\ell_2$-norm radius $r$ centered at $\alpha$. Further construct $\Phi_\epsilon$ using $\mathcal{B}_r$ as follows:

$$\Phi_\epsilon(x) = \begin{cases} \begin{bmatrix} z_c \\ 0 \end{bmatrix}, & z_e \in \mathcal{B}_r \\ \begin{bmatrix} z_c \\ z_e \end{bmatrix}, & z_e \in \mathcal{B}_r^c, \end{cases} \quad \text{and} \quad \hat{\beta} = \begin{bmatrix} \beta_c \\ \hat{\beta}_e \\ \beta_0 \end{bmatrix}. \tag{19}$$

Without loss of generality, fix an environment $e$.

1. By Lemma D.1, the squared gradient norm is upper bounded by

$$\|\nabla_{\hat{\beta}} \mathcal{R}^e(\Phi_\epsilon, \hat{\beta})\|_2^2 \leq \left\| \mathbb{P}(z_e \in \mathcal{B}_r^c) \mathbb{E}[|z_e| \mid z_e \in \mathcal{B}_r^c] \right\|_2^2. \tag{20}$$

Define $B_e := B_r(\mu_e)$, and observe that $\mathcal{B}_r^c \subseteq B_e^c$. Since $|z_e|$ is non-negative,

$$\mathbb{P}(z_e \in \mathcal{B}_r^c) \mathbb{E}[|z_e| \mid z_e \in \mathcal{B}_r^c] \leq \mathbb{P}(z_e \in B_e^c) \mathbb{E}[|z_e| \mid z_e \in B_e^c]$$

(this inequality is element-wise). Plugging this into Equation 20 yields

$$\|\nabla_{\hat{\beta}} \mathcal{R}^e(\Phi_\epsilon, \hat{\beta})\|_2^2 \leq \left\| \mathbb{P}(z_e \in B_e^c) \mathbb{E}[|z_e| \mid z_e \in B_e^c] \right\|_2^2$$

$$= [\mathbb{P}(z_e \in B_e^c)]^2 \left\| \mathbb{E}[|z_e| \mid z_e \in B_e^c] \right\|_2^2.$$

Define $p = \mathbb{P}(z_e \in \mathcal{B}_r^c) \leq \mathbb{P}(z_e \in B_e^c)$. By Lemma F.3,

$$p \leq p_{\epsilon,d_e} = e^{-d_e \min((\epsilon-1),(\epsilon-1)^2)/8}.$$

Combining Lemmas F.4 and F.5 gives

$$\left\| \mathbb{E}[|z_e| \mid z_e \in B_e^c] \right\|_2^2 \leq 2d_e \left[ \sigma \frac{\phi(r/\sqrt{d_e})}{F(-r/\sqrt{d})} \right]^2 + 2\|\mu_e\|_2^2$$

$$\leq d_e \sigma_e^2 \exp\left\{ 2\epsilon(\sigma_e^2 - 1/2) \right\} \left[ \epsilon \sigma_e^2 + 1 \right] + 2\|\mu_e\|_2^2.$$

Putting these two bounds together, we have

$$\|\nabla_{\hat{\beta}} \mathcal{R}^e(\Phi_\epsilon, \hat{\beta})\|_2^2 \in \mathcal{O}\left( p_{\epsilon,d_e}^2 \left[ \epsilon d_e \sigma_{\max}^4 \exp\{2\epsilon\sigma_{\max}^2\} + \|\mu_e\|_2^2 \right] \right),$$

and averaging this value across environments gives the result.

2. $\Phi_\epsilon, \hat{\beta}$ is equal to the optimal invariant predictor on $\mathcal{B}_r$ and the ERM solution on $\mathcal{B}_r^c$. The claim then follows from Lemma F.3.

3. This follows directly from Lemma D.2.

4. With Equation 16, we have that

$$\beta_{e;\text{ERM}}^T \mu_{E+1} = -\sum_{e \in \mathcal{E}} \alpha_e \beta_{e;\text{ERM}}^T \mu_e$$

$$\leq -2(1-\gamma) \sum_{e \in \mathcal{E}} \alpha_e \frac{\|\mu_e\|_2^2}{\sigma_e^2}$$

$$\leq -2(1-\gamma) \frac{\|\mu_c\|_2^2/\sigma_c^2 + |\beta_0|/2 + c\sigma_{\text{ERM}}}{1-\gamma}$$

$$= -(2\|\mu_c\|_2^2/\sigma_c^2 + |\beta_0| + 2c\sigma_{\text{ERM}}).$$

where we have applied Assumption 1 in the first inequality and Equation 17 in the second. Consider the full set of features $\Phi_\epsilon(x) = f^{-1}(x)$, and without loss of generality assume $y = 1$. The label-conditional distribution of the resulting logit is

$$\beta_c^T z_c + \beta_{e;\text{ERM}}^T z_e + \beta_0 \sim \mathcal{N}\left(\beta_c^T \mu_c + \beta_{e;\text{ERM}}^T \mu_{E+1} + \beta_0, \sigma_{\text{ERM}}^2\right).$$

Therefore, the 0-1 risk is equal to the probability that this logit is negative. This is precisely

$$F\left(-\frac{\beta_c^T \mu_c + \beta_{e;\text{ERM}}^T \mu_{E+1} + \beta_0}{\sigma_{\text{ERM}}}\right) \geq F\left(\frac{(2\|\mu_c\|_2^2/\sigma_c^2 + |\beta_0| + 2c\sigma_{\text{ERM}}) - 2\|\mu_c\|_2^2/\sigma_c^2 - |\beta_0|}{\sigma_{\text{ERM}}}\right)$$

$$= F\left(\frac{2c\sigma_{\text{ERM}}}{\sigma_{\text{ERM}}}\right)$$

$$= F(2c).$$

Observe that by the previous part, $\Phi_\epsilon \neq f^{-1}$ on at most a q fraction of observations, so the risk of our predictor $\Phi_\epsilon, \hat{\beta}$ can differ from that of $f^{-1}, \hat{\beta}$ by at most $q$. Therefore our predictor's risk is lower bounded by $F(2c) - q$. In particular, choosing $c = 1$ recovers the statement in the main body. $\qquad\square$

## D.2 Discussion of Conditions and Assumption

To see just how often we can expect the conditions for Theorem D.3 to hold, we can do a rough approximation based on the expectations of each of the terms. A reasonable prior for the environmental means is a multivariate Gaussian $\mathcal{N}(m, \Sigma)$. We might expect them to be very concentrated (with $\text{Tr}(\Sigma)$ small), or perhaps to have a strong bias (with $\|m\|_2^2 \gg \text{Tr}(\Sigma)$). For simplicity we treat the variances $\sigma_c^2, \sigma_e^2$ as constants. Then the expected separation between any two means from this distribution is

$$\mathbb{E}[\|\mu_1 - \mu_2\|_2] = \mathbb{E}_{x \sim \mathcal{N}(0,2\Sigma)}[\|x\|_2] \approx \sqrt{2\,\text{Tr}(\Sigma)}.$$

In high dimensions this value will tightly concentrate around the mean, which is in $\mathcal{O}(\sqrt{d_e})$. On the other hand, even a slight deviation from this separation, to $\Omega(\sqrt{d_e \log E})$, means $\delta \in \Omega(\sqrt{\log E})$, which implies $q \in \mathcal{O}(1/E)$; this is plenty small to ensure worse-than-random error on the test distribution.

Now we turn our attention to the second condition (17). The expected squared norm of each mean is $d_e$, and in high dimensions we expect them to be reasonably orthogonal (as a rough approximation; this is technically not true with a non-centered Gaussian). Then so long as $\sum_i \alpha_i \in \Omega(1)$, the left-hand side of Equation 17 is approximately $d_e$. On the other hand, treating $\gamma$ as a constant, the right-hand side is close to $d_c + \sqrt{d_c + d_e} \in \mathcal{O}(d_c + \sqrt{d_e})$. Thus, Equation 17 is quite likely to hold for any mean $\mu_{E+1}$ with the same scale as the training environments but with reversed correlations—again, this is *exactly the situation* where IRM hopes to outperform ERM, and we have shown that it does not.

We can also do a quick analysis of Assumption 1 under this prior: the ERM-optimal non-invariant coefficient will be approximately $2m/\sigma_e^2$ with high probability, meaning $\hat{\beta}^T \mu \approx 2\|m\|_2^2/\sigma_e^2$ for every environment. Thus, this vector will be $\gamma$-close to optimal with $\gamma \approx 0$ for every environment with high probability.

## E Extensions to Alternative Objectives

### E.1 Extensions for the Linear Case

Observe that the constraint of Equation 4 is strictly stronger than that of Bellot & van der Schaar (2020); when the former is satisfied, the penalty term of the latter is necessarily 0. It is thus trivial to extend all results in the Section 5 to this objective. As another example, consider the risk-variance-penalized objective of Krueger et al. (2020):

$$\min_{\Phi, \hat{\beta}} \quad \frac{1}{|\mathcal{E}|} \sum_{e \in \mathcal{E}} \mathcal{R}^e(\Phi, \hat{\beta}) + \lambda \text{Var}_{e \in \mathcal{E}}\left(\mathcal{R}^e(\Phi, \hat{\beta})\right), \tag{21}$$

It is simple to extend Theorem 5.1 under an additional assumption:

**Corollary E.1** (Extension to Theorem 5.1). *Assume $f$ is linear. Suppose we observe $E \le d_e$ environments with linearly independent means and identical variance $\sigma_e^2$. Consider minimizing empirical risk subject to a penalty on the risk variance (21). Then there exists a $\Phi, \hat{\beta}$ dependent on the non-invariant features which achieves a lower objective value than the optimal invariant predictor for* any *choice of regularization parameter $\lambda \in [0, \infty]$.*

*Proof.* Consider the featurizer $\Phi$ constructed in Lemma C.2. If the environmental variance is constant, then the label-conditional distribution of the environmental features,

$$z_e \mid y \sim \mathcal{N}(y \cdot \tilde{\mu}\sigma_e^2, \sigma_e^2),$$

is also invariant. This implies that the optimal $\hat{\beta}$ also has constant risk across the environments, meaning the penalty term is 0, and as a result the objective does not depend on the choice of $\lambda$. As in 5.1, invoking Lemma F.1 implies that the overall risk is lower than that of the optimal invariant predictor. $\square$

As mentioned in Section 5, this additional requirement of constant variance is due to the assumptions underlying the design of the objective—REx expects additional invariance of the second moment $\text{Var}(y \mid \Phi(x))$, in contrast with the strictly weaker invariance of $\mathbb{E}[y \mid \Phi(x)]$ assumed by IRM. This might seem to imply that REx is a more robust objective, but this does not convey the entire picture. The conditions for the above corollary are just one possible failure case for REx; by extending Theorem 5.3 to this objective, we see that REx is just as prone to bad solutions:

**Corollary E.2** (Extension to Theorem 5.3). *Suppose we observe $E \le d_e$ environments, such that all environmental means are linearly independent. Then there exists a $\Phi, \hat{\beta}$ which uses* only *environmental features and, under any choice of $\lambda \in [0, \infty]$, achieves a lower objective value than the optimal invariant predictor under 0-1 loss on every environment $e$ such that $\tilde{\mu} > \sigma_c^{-1}\|\mu_c\|_2 + \frac{|\beta_0|}{2\sigma_c^{-1}\|\mu_c\|_2}$.*

*Proof.* We follow the proof of Theorem 5.3, except when solving for $p$ as in Lemma C.1 we instead find the unit-norm vector such that

$$p^T \mu_e = \sigma_e \tilde{\mu} \quad \forall e \in \mathcal{E}. \tag{22}$$

Observe that by setting $\Phi(x) = [p^T z_e]$ and $\hat{\beta} = [1]$, the 0-1 risk in a given environment is

$$\eta F(-\tilde{\mu}\sigma_e/\sigma_e) + (1 - \eta)F(-\tilde{\mu}\sigma_e/\sigma_e) = F(-\tilde{\mu}),$$

which is independent of the environment. Further, by carrying through the same proof as in Theorem 5.3, we get that this non-invariant predictor has lower 0-1 risk so long as

$$\alpha + \frac{|\beta_0|}{2\alpha} \le \tilde{\mu},$$

where $\alpha = \sigma_c^{-1}\|\mu_c\|_2$ $\square$

Though $\tilde{\mu}$ here is not exactly the same value because of the slightly different solution (22), it depends upon the geometry of the training environments in the same way—it is the same as taking the square root of each of the variances. We can therefore expect this condition to hold in approximately the same situations, which we empirically verify by replicating Figure C.2 with the modified equation below.

## E.2 EXTENSIONS FOR THE NON-LINEAR CASE

The failure of these objectives in the non-linear regime is even more straightforward, as we can keep unchanged the constructed predictor from Theorem 6.1. Observe that parts 2-4 of the theorem do not involve the objective itself, and therefore do not require modification.

To see that part 1 still holds, note that since the constructed predictor matches the optimal invariant predictor on $1 - p$ of the observations, its risk across environments can only vary on the remaining $p$ fraction: thus the centered 0-1 risk is bounded between 0 and $p$. It is immediate that the variance of the environmental risks is upper bounded by $\frac{p^2}{4} \in \mathcal{O}(p^2)$. Applying this argument to the other objectives yields similar results.

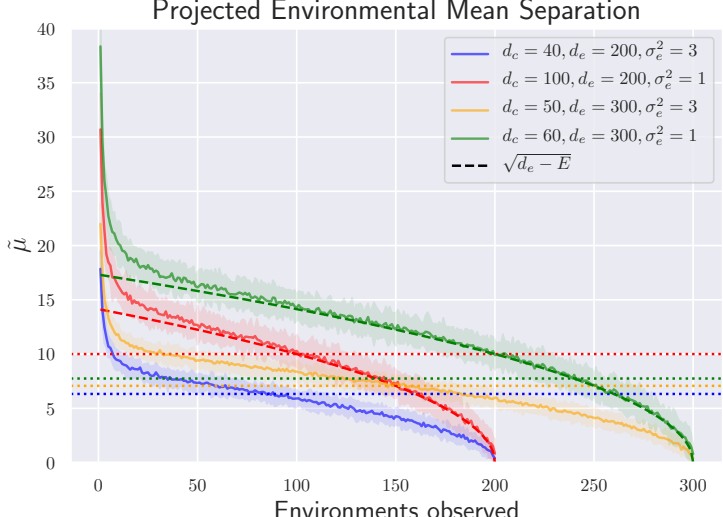

Figure E.1: Simulations to evaluate $\tilde{\mu}$ for varying ratios of $\frac{d_e}{d_c}$. When $\sigma_e^2 = 1$ the value closely tracks $\sqrt{d_e - E}$, and the crossover point is approximately $d_e - \sigma_e^2 d_c$. Due to the similarity of Equation 22 to Equation 11, it makes sense that the results are very similar to those presented in Figure C.2.

## F   TECHNICAL LEMMAS

**Lemma F.1.** *Consider solving the standard logistic regression problem*

$$z \sim p(z) \in \mathbb{R}^k,$$
$$y = \begin{cases} +1 & w.p. \ \sigma(\beta^T z), \\ -1 & w.p. \ \sigma(-\beta^T z). \end{cases}$$

*Assume that none of the latent dimensions are degenerate—$\forall S \subseteq [k]$, $\mathbb{P}(\beta_S^T z_S \neq 0) > 0$, and no feature can be written as a linear combination of the other features. Then for any distribution $p(z)$, any classifier $f(z) = \sigma(\beta_S^T z_S)$ that uses a strict subset of the features $S \subsetneq [k]$ has strictly higher risk with logistic loss than the Bayes classifier $f^*(z) = \sigma(\beta^T z)$. This additionally holds for 0-1 loss if $\beta_{-S}^T z_{-S}$ has greater magnitude and opposite sign of $\beta_S^T z_S$ with non-zero probability.*

*Proof.* The Bayes classifier suffers the minimal expected loss for each observation z. Therefore, another classifier has positive excess risk if and only if it disagrees with the Bayes classifier on a set of non-zero measure. Consider the set of values $z_{-S}$ such that $\beta_{-S}^T z_{-S} \neq 0$. Then on this set we have

$$f^*(\beta^T z) = \sigma(\beta_S^T z_S + \beta_{-S}^T z_{-S}) \neq \sigma(\beta_S^T z_S) = f(z).$$

Since these values occur with positive probability, $f$ has strictly higher logistic risk than $f^*$. By the same argument, there exists a set of positive measure under which

$$f^*(\beta^T z) = \text{sign}(\beta_S^T z_S + \beta_{-S}^T z_{-S}) \neq \text{sign}(\beta_S^T z_S) = f(z),$$

and so $f$ also has strictly higher 0-1 risk. □

**Lemma F.2.** *For any feature vector which is a linear function of the invariant and environmental features $\tilde{z} = Az_c + Bz_e$, any optimal corresponding coefficient for an environment $e$ is of the form*

$$2(AA^T \sigma_c^2 + BB^T \sigma_e^2)^+ (A\mu_c + B\mu_e),$$

*where $G^+$ is a generalized inverse of $G$.*

*Proof.* We begin by evaluating a closed form for $p^e(y \mid \tilde{z})$. We have:

$$p^e(y \mid Az_c + Bz_e = \tilde{z})$$

$$= \frac{p(Az_c + Bz_e = \tilde{z} \mid y)p(y)}{p^e(Az_c + Bz_e = \tilde{z})}$$

$$= \frac{p^e(Az_c + Bz_e = \tilde{z} \mid y)}{p^e(Az_c + Bz_e = \tilde{z} \mid y) + p^e(Az_c + Bz_e = \tilde{z} \mid -y)}$$

$$= \frac{1}{1 + \frac{p^e(Az_c + Bz_e = \tilde{z} \mid -y)}{p^e(Az_c + Bz_e = \tilde{z} \mid y)}}.$$

Now we need a closed form expression for $p(Az_c + Bz_e = \tilde{z} \mid y)$. Noting that $z_c \perp\!\!\!\perp z_e \mid y$, this is a convolution of the two independent Gaussian densities, which is the density of their sum. In other words,

$$Az_c + Bz_e \mid y \sim \mathcal{N}(y \cdot (\overbrace{A\mu_c + B\mu_e}^{\bar{\mu}}), \overbrace{AA^T\sigma_c^2 + BB^T\sigma_e^2}^{\bar{\Sigma}}).$$

Thus,

$$p^e(Az_c + Bz_e = \tilde{z} \mid y) = \frac{1}{\left(2\pi|\bar{\Sigma}|\right)^{k/2}} \exp\left\{-\frac{1}{2}(\tilde{z} - y \cdot \bar{\mu})^T \bar{\Sigma}^+ (\tilde{z} - y \cdot \bar{\mu})\right\}.$$

Canceling common terms, we get

$$p^e(y = 1 \mid Az_c + Bz_e = \tilde{z}) = \frac{1}{1 + \frac{p^e(Az_c + Bz_e = \tilde{z} \mid -y)}{p^e(Az_c + Bz_e = \tilde{z} \mid y)}}$$

$$= \frac{1}{1 + \exp\left\{-y \cdot 2\tilde{z}^T \bar{\Sigma}^+ \bar{\mu}\right\}}$$

$$= \sigma\left(y \cdot 2\tilde{z}^T \bar{\Sigma}^+ \bar{\mu}\right).$$

Therefore, given a feature vector $\tilde{z}$, the optimal coefficient vector is $2\bar{\Sigma}^+\bar{\mu}$. $\qquad\square$

**Lemma F.3.** *For any environment $e$ with parameters $\mu_e, \sigma_e^2$ and any $\epsilon > 1$, define*

$$B := B_{\sqrt{\epsilon\sigma_e^2 d_e}}(\mu_e),$$

*where $B_r(\alpha)$ is the ball of $\ell_2$-norm radius $r$ centered at $\alpha$. Then for an observation drawn from $p^e$, we have*

$$\mathbb{P}_{z_e \sim p^e}(z_e \in B^c) \leq \exp\left\{-\frac{d_e \min((\epsilon - 1), (\epsilon - 1)^2)}{8}\right\}.$$

*Proof.* Without loss of generality, suppose $y = 1$. We have

$$\mathbb{P}(z_e \in B) \geq \mathbb{P}_{z_e \sim \mathcal{N}(\mu_e, \sigma_e^2 I)}\left(\|z_e - \mu_e\|_2 \leq \sqrt{\epsilon\sigma_e^2 d_e}\right)$$

$$= \mathbb{P}_{z_e \sim \mathcal{N}(0, \sigma_e^2 I)}\left(\|z_e\|_2 \leq \sqrt{\epsilon\sigma_e^2 d_e}\right)$$

$$= \mathbb{P}_{z_e \sim \mathcal{N}(0, I)}\left(\|z_e\|_2^2 \leq \epsilon d_e\right).$$

Each term in the squared norm of $z_e$ is a random variable with distribution $\chi_1^2$, which means their sum has mean $d_e$ and is sub-exponential with parameters $(2\sqrt{d_e}, 4)$. By standard sub-exponential concentration bounds we have

$$\mathbb{P}_{z_e \sim \mathcal{N}(0, I)}\left(\|z_e\|_2^2 \geq \epsilon d_e\right) \leq \exp\left\{-\frac{d_e \min((\epsilon - 1), (\epsilon - 1)^2)}{8}\right\},$$

which immediately implies the claim. $\qquad\square$

**Lemma F.4.** *Let $z \sim \mathcal{N}(\mu, \sigma^2 I_d)$ be a multivariate isotropic Gaussian in $d$ dimensions, and for some $r > 0$ define $B$ as the ball of $\ell_2$ radius $r$ centered at $\mu$. Then we have*

$$\left\| \mathbb{E}[|z| \mid z \in B^c] \right\|_2^2 \leq 2d \left[ \sigma \frac{\phi(r/\sqrt{d})}{F(-r/\sqrt{d})} \right]^2 + 2\|\mu\|_2^2,$$

*where $\phi, F$ are the standard Gaussian PDF and CDF.*

*Proof.* Observe that

$$\mathbb{E}_{z \sim \mathcal{N}(\mu, \sigma^2 I_d)}[|z| \mid z \in B^c] = \mathbb{E}_{z \sim \mathcal{N}(\mu, \sigma^2 I_d)}[|z| \mid \|z - \mu\|_2 > r]$$

$$= \mathbb{E}_{z \sim \mathcal{N}(0, \sigma^2 I_d)}[|z + \mu| \mid \|z\|_2 > r]$$

$$\leq \mathbb{E}_{z \sim \mathcal{N}(0, \sigma^2 I_d)}[|z| \mid \|z\|_2 > r] + |\mu|.$$

Now, consider the expectation for an individual dimension, and note that $|z_i| > \frac{r}{\sqrt{d}} \ \forall i \implies \|z\|_2 > r$. So because the dimensions are independent, conditioning on this event can only increase the expectation:

$$\mathbb{E}_{z \sim \mathcal{N}(0, \sigma^2 I_d)}[|z_i| \mid \|z\|_2 > r] \leq \mathbb{E}_{z_i \sim \mathcal{N}(0, \sigma^2)} \left[ |z_i| \mid |z_i| > \frac{r}{\sqrt{d}} \right]$$

$$= \mathbb{E}_{z_i \sim \mathcal{N}(0, \sigma^2)} \left[ z_i \mid z_i > \frac{r}{\sqrt{d}} \right],$$

where the equality is because the distribution is symmetric about 0. This last term is known as the *conditional tail expectation* of a Gaussian and is available in closed form:

$$\mathbb{E}_{z_i \sim \mathcal{N}(0, \sigma^2)} \left[ z_i \mid z_i > \frac{r}{\sqrt{d}} \right] = \sigma \frac{\phi(F^{-1}(\alpha))}{1 - \alpha},$$

where $\alpha = F(r/\sqrt{d})$. Combining the above results, squaring with $(a + b)^2 \leq 2(a^2 + b^2)$, and summing over dimensions, we get

$$\left\| \mathbb{E}[|z| \mid z \in B^c] \right\|_2^2 \leq 2 \sum_{i=1}^{d} \mathbb{E}_{z_i \sim \mathcal{N}(0, \sigma^2)} \left[ z_i \mid z_i > \frac{r}{\sqrt{d}} \right]^2 + 2\|\mu\|_2^2$$

$$= 2d \left[ \sigma \frac{\phi(r/\sqrt{d})}{F(-r/\sqrt{d})} \right]^2 + 2\|\mu\|_2^2,$$

as desired. $\square$

**Lemma F.5.** *For $\sigma, \epsilon > 0$, define $r = \sqrt{\epsilon}\sigma$. Then*

$$\left[ \frac{\phi(r)}{F(-r)} \right]^2 \leq \frac{1}{2} \exp\left\{ 2\epsilon(\sigma^2 - 1/2) \right\} \left[ \epsilon\sigma^2 + 1 \right].$$

*Proof.* We have

$$\phi(r) = \frac{1}{\sqrt{2\pi}} \exp\left\{ -\frac{\epsilon}{2} \right\}$$

and

$$F(-r) \geq \frac{2 \exp\{-\epsilon\sigma^2\}}{\sqrt{\pi}(\sqrt{\epsilon}\sigma + \sqrt{\epsilon\sigma^2 + 2})}$$

(see Kschischang (2017)). Dividing them gives

$$\frac{\phi(r)}{F(-r)} \leq \frac{1}{2\sqrt{2}} \exp\{\epsilon(\sigma^2 - 1/2)\} \left[ \sqrt{\epsilon}\sigma + \sqrt{\epsilon\sigma^2 + 2} \right].$$

Squaring and using $(a + b)^2 \leq 2(a^2 + b^2)$ yields the result. $\square$

