# OpenReview forum: "The Risks of Invariant Risk Minimization"
_ICLR.cc/2021/Conference — ICLR 2021 Poster_

### Official Review · AnonReviewer1 · 2020-10-24
**Interesting results, but I believe slightly misleading**

**Rating:** 6
**Confidence:** 4

**Review:**

The paper, entitled "The Risks of Invariant Risk Minimization", provides a theoretical analysis of the IRM learning objective under particular conditions. It claims to provide two theoretical results, namely:
1) in the linear setting (i.e., the optimal model is linear), they provide a necessary and sufficient condition for IRM to recover the optimal invariant predictor.
2) in the non-linear setting, they demonstrate that IRM does not perform better than ERM (empirical risk minimization).

Strong points of the paper:
1) the paper addresses an important question: what are the theoretical guarantees of the IRM objective ?
2) the paper provides a solid argument against the IRM objective, which is shown to be no better than ERM, in a limited but non-trivial setting
3) the paper tries to relate IRM to several prior works from the literature

Weak points of the paper:
1) I believe the paper contains a major flaw: the so-called "invariant classifier", which uses only invariant features, is not the optimal invariant predictor sought by IRM. As such the first claim of the authors, necessary and sufficient conditions under which IRM recovers the optimal invariant predictor in the linear setting, is wrong.
2) The paper suffers from a poor structure, a lack of clarity, and a general lack of rigor in the definition of the key concepts.

Despite commendable efforts from the authors to study IRM theoretically, even in a limited setting, I recommend rejection for the paper.

First, while it appears rather theoretical, the paper lacks rigor and clarity in the definition of the problem, the key concepts, and the assumptions. The description of the OOD generalization problem, of IRM and its assumptions, of the assumptions made by the authors in their simplified model, and of the introduced concepts (notably the "invariant classifier"), are lacking a formal mathematical description. The main results of the paper are given in Section 3, before the studied object, IRM, is even formally  introduced in Section 4. The authors interchangeably use the adjectives "causal" or "invariant", without having properly defined the meaning of those concepts in the first place, and with no particular utility. Maybe IRM can be given a causal interpretation, under some assumptions, but clearly this is out of the scope of the paper. The theoretical analysis of IRM conducted in the paper does not require the notion of causality, which only brings confusion to the paper in my opinion.

Second, and most importantly, I believe that one of the author's claims (which occupies about half the paper) is simply wrong (see my detailed comments below with a counter-example). There seems to be a misunderstanding as to what IRM actually seeks for, which is not the so-called "invariant classifier" that uses only invariant features. IRM does not seek to use invariant features only, but to learn those invariant features, possibly by using non-invariant ones. As such, results such as Theorem 5.1, Corollary 5.2, and Theorem 5.3, have very little value I think. The second claim of the authors remains interesting, however I believe the paper requires a substantial revision and in its current state can not be accepted at ICLR.

I would appreciate if the authors could formally describe what they mean by "invariant classifier", and point me to the evidence where the IRM authors argue that IRM will recover such a classifier.

**Post-discussions**: the authors have clarified the meaning of "invariant classifier", which is now tied to their specific toy problem, and I now believe such a classifier is indeed the minimax OOD classifier supposedly seeked by IRM. While I still think the paper does not shine on the side of clarity, my main concern about the incorrectness of the presented theorems has been answered and I believe the paper will be of interest to the community. I therefore raise my recommendation towards acceptance.

Below are my detailed comments:

p.1 §3: each environment corresponds to interventions on the SEM (Pearl, 2009) on just the non-causal mechanisms -> What does non-causal mechanism mean here ? If the SEM is given a causal interpretation, then all mechanisms are causal. Do you mean the mechanisms which are not causaly impacting the variable of interest Y ?

p.1 §3: invariant features in the SEM -> Do you mean the causal features ? Or the invariant mechanisms ? Invariant features does not make sense. If interpreted as constant features, those are uninformative. If interpreted as having the same distribution across environments, this does not make sense either, since it does not guarantee such a feature will be causal.

p.2 §5:  invariance of the feature-conditioned label distribution -> I like the effort the authors put in grouping the existing works in a unified framework and notation. There is however a bit of inconsistency and ambiguity in the terminologies employed throughout the paper. Invariant distribution? Domain-invariant representation ? Invariant features ? A proper definition would help, and enforcing consistency throughout the paper, by using the same names when those refer to the same concepts, would benefit readability.

p.2 §5: The main difference [...] train to test distributions. -> I like that the authors make this comparison, but I also would have liked this discussion to be further extended. There is a clear relationship indeed. If IRM looks for a representation $\phi(x)$ such that $p(\phi(x))$ can change between domains, but $p(y|\phi(x))$ remains the same, then this is exactly the assumption of covariate shift: $p(x)$ can change, but $p(y|x)$ remains the same. However, finding such a representation $\phi$ does not entirely solve the problem. First, the representation has to be predictive enough for $y$, ideally so that $y \perp x | \phi(x)$. Otherwise, even a constant $\phi$ might result in $p(y|\phi(x))$ being invariant if $p(y)$ is. But then marginal predictions from $p(y)$ would probably be sub-optimal. Second, while in theory (infinite model capacity, infinite data) covariate shift is not an issue (Bayes-optimal predictors do not need $p(x)$, only $p(y|x)$ over the support of $x$), in practice it is another story, and additional methods such as sample reweighting can be beneficial. See paragraphs 3,4 from the introduction in V. Tran, and A. Aussem. A Practical Approach to Reduce the Learning Bias Under Covariate Shift. ECML/PKDD 2015.

p.2 §6: whose joint distribution with the label is fixed for all environments -> this is a strong assumption which IRM, as far as I understand, does not make

p.3 §3: invariant (causal) relationship -> I am a bit uncomfortable with the authors mixing the invariant and causal adjectives in this context. In your SCM, if given a causal interpretation, $y$ is a cause of $z_c$, and $p(y|z_c) \neq p(y|do(z_c))$. I would suggest for the authors to stick to the concept of invariance (which they still have not defined properly...), and drop the causal interpretation, which brings nothing but confusion to the paper. Causality really is a different business, which is not required to study the capabilities of the IRM objective.

p.3 §4: the causation can just as easily be viewed in the other direction -> Exactly. So why even talking about causality ? Do you want a model of $p(y|x)$, which generalize across different environments with different distributions for p(x) ? Or do you want a causal model of $p(y|do(x))$ ? Those are different objectives, and using them interchangeable really is confusing, as they do not require the same assumptions. Here, as you say, we do not care whether $x$ causes $y$ or $y$ causes $x$.

p.3 §5: a constant marginal -> Marginal of what ? Please be precise.

p.3 §5: is necessary: Necessary to what ? Which assumptions of IRM would be violated as a result of not having $y \perp e$ your model ? The IRM assumptions were never clearly stated. Would that make finding $\psi$ such that $p(y|\psi(x))$ is invariant impossible ?

p.3 §8: whose joint or conditional distribution -> The conditional is implied by the joint. You can remove "or conditional" here.

p.3 §8: Definition 1 -> The paper would gain a lot in clarity if this "invariant classifier" was given a formal, mathematical definition. For example, what do you mean by "invariant features" ? The same holds for you assumptions, which could be defined more clearly: $z_c,y \perp e$, $e$ uniform. According to your definition, I can think of the following:

$$
h^\star = \arg \min_h \mathbb{E}_{(x,y,z_c,z_e,e)}[l(h(x), y)], \text{subject to } h(f(z_c,z_e)) = h(f(z_c,z'_e)), \forall z_c,z_e,z'_e \text{,}
$$

The "invariant classifier" from this definition is not the one that IRM seeks. It suffices to consider a scenario where $f=I$ $Z_c=\emptyset$, and two environments such that $\mu_{e_1}, \sigma_{e_1}=(100, 1)$ and $\mu_{e_2}, \sigma_{e_2}=(1000, 1)$. Then $z_e$ is not an invariant feature, but clearly the representation $f(z_e)=sign(z_e)$ is invariant, and is very informative of $y$. I think this is a major confusion in the article, which becomes clear only when ones writes down this definition formally.

p.4 §1: Observe that [...] the invariant classifier does not. -> This is I believe the major flaw of the article. The objective in OOD generalization is to find the Bayes classifier, not the "invariant classifier". Likewise, the objective of IRM is not to learn the "invariant classifier". As such, showing that IRM does not recover this "invariant classifier" has little value.

p.4 §3: Since those are informal theorems, I would rather name them differently, e.g.,  Result 1, Result 2, Result 3.

p.4 Section 4: The definition of IRM should be presented before the Section 3: results.

p.4 Section 4 §2: The authors argue that such a function will use only invariant features -> I do not find this argument in the original IRM paper.

p.5 Theorem 5: E training environments -> and how many environments in the test distribution ?

p.5 Appendic C.1: In Appendix Figure C.1, I believe the y axis should read $\eta$, not Accuracy ? Also, I fail to understand why ERM is a flat line in this Figure. If more environments are provided, even with unlimited data, then the ERM classifier should change, and the test performance as well. Maybe I am missing something here, but the experimental setup (such as how model parameters are chosen for each new environment) is not clearly described.

p.5 Corollary 5.2: I am not sure what this implies. Maybe IRM does not recover the "invariant classifier" in that case, but what we really want for OOD generalization is the Bayes-optimal classifier. Again, it is not clearly stated what IRM wants to achieve...

p.6 Theorem 5.3: The initial assumptions these results rely on should be stated formally. Do these results hold for any $p(x,y,e)=p(e)p(y)p(x|e,y)$ distribution ? For any $p(x,y,e)$ distribution satisfying the IRM assumptions ? Only for the specific model the authors consider ?

p.6 Theorem 5.3: feasible -> Feasible for IRM ?

p.6 Figure C.2: Some labels are missing from Figure C.2. What do the doted horizontal lines represent ?

p.8 Conclusion: the number of training environments needed to ensure good generalization under arbitrary distribution shift is linear in the number of non-invariant features -> I believe this is wrong. The Bayes classifier is the one ensuring good OOD generalization. The "invariant classifier", which you prove requires a number of environments equal to the number of non-invariant features, is not the Bayes classifier.

---

> ### Author Response · Authors · 2020-11-14
> **Thanks for reading so thoroughly, but you've misunderstood IRM (1/3)**
>
> Thank you for taking the time to review this work so thoroughly! Your review expresses two major concerns:
> 1) Our results are incorrect and not adequately formal.
> 2) The use of the word ‘causal’ and all related terminology.
>
> We will respond to these points first, then subsequently the more detailed points.
>
> Regarding the second point: we don’t use causality terminology in the formal parts of the paper, merely in the introductory, informal portions. In fact, the original IRM paper (Arjovsky et al.) copiously uses causal nomenclature—a quick “search” returns 77 hits—despite (as you yourself noted) nothing in the IRM formulation explicitly leveraging causality. We discuss this in more detail further down in our response.
>
> **With regards to the first point:**
> $\textbf{We are }\ \textit{absolutely certain}\ \ \textbf{that this concern is due to a fundamental misunderstanding of the}$$\textbf{premise and objective of both the original IRM paper and this work. We hope our detailed response below will}$$\textbf{demonstrate to you that our results are both correct and meaningful.}$
> We further hope that with your help, we can understand more precisely the source of this misunderstanding, which will help us rephrase parts of the work to preempt this confusion for future readers.
>
> ---
>
> We apologize if you found our setup difficult to understand, though we note that this sentiment was not shared by the other reviewers—reviewers 2 and 4 explicitly noted an appreciation for the care we took to make the work intuitive and easy to follow. In general, especially in the introductory parts, we deliberately use expository language rather than formal math for ease of reading. This is why we have informal theorem statements first, which we make fully formal later; we feel this helps readers who are not experts in this subarea. We don’t think our definitions are particularly informal; we think rather a major source of confusion is the distinction between the Bayes and invariant classifiers. We will make sure to spell this out more forcefully in the next version. In the meantime, we will clarify the distinction in this rebuttal to hopefully rectify the misunderstanding.
>
> You write that “the objective in OOD generalization is to find the Bayes classifier, not the "invariant classifier".” We assume what you mean by this is: “the goal of OOD generalization is to learn a classifier which is Bayes for a new test environment”. **In the context of this paper, and also that of (Peters et al., Arjovsky et al.), this statement is not correct**. To clarify: the Bayes classifier in a new environment would make use of all available information in $x$, while the invariant classifier makes use of only “invariant features”. Note in Arjovsky et al. there is no formal definition of “invariant” or “causal” features as the paper doesn’t assume a data model, and the objective itself doesn’t come with formal guarantees with respect to the data distribution. In fact, part of the motivation for our paper was to propose a data model which formalizes some of the intuitions behind the IRM objective and to show how solving it in fact does *not* match this intuition. Our data model, **completely described** by eqs (1)-(3), explicitly defines which features are “invariant” and which are not (surely these equations could not be considered informal). Furthermore, we think it accurately captures the intuition of Arjovsky et al.: the label-conditional distribution of the "invariant" features is constant across environments, whereas the label-conditional distribution of the "non-invariant" features can vary; **these definitions are provided at the beginning of Section 3**. Note, our setup *explicitly allows* for the existence of features which are informative of the label yet non-invariant; the invariant classifier *intentionally ignores* these features. This is the ultimate objective of IRM.

---

> > ### Author Response · Authors · 2020-11-14
> > **Thanks for reading so thoroughly, but you've misunderstood IRM (2/3)**
> >
> > To see why the Bayes classifier is *not our objective*, consider classifying data from our proposed model. For latents $(z_c, z_e)$, the Bayes classifier will make the risk-minimizing prediction $\mathbb{E}[y | z_c, z_e] = \sigma(\beta_c^Tz_c + \beta_e^Tz_e + \beta_0)$, where $\beta_e = \frac{2\mu_e}{\sigma_e^2}$ and $\beta_c, \beta_0$ are defined as in the equation after Definition 1. Observe that this prediction uses all available information, including the environmental features $z_e$. However, *this predictor is not constant across all environments*: $\beta_e$ will by definition be different for environments with different means and variances. As a result, when considering the full set of latents $(z_c, z_e)$, there is no *fixed* linear classifier which is simultaneously Bayes for all environments. Note the connection here to the IRM objective: the goal is to learn a featurizer such that a single classifier is optimal (i.e., Bayes) for *each environment separately and simultaneously*. Since this is not possible when the featurizer extracts $z_e$, the IRM objective enforces learning a feature extractor that *ignores* $z_e$. So it is clear that the Bayes classifier is not our objective—**it is not even in our hypothesis class!**
> >
> > Returning to IRM, it is essential to understand *why* the goal is to ignore such features. **Here we address the flaw in your counter-example.** You are correct that $\Phi(f(z_c, z_e)) = \textrm{sign}(z_e)$ is very informative of the label $y$, but *the way in which it is informative is not constant*. The key point is that in future test environments, the environmental feature distribution could be *arbitrarily* different; for example, consider the test environment $(\mu_e, \sigma_e^2) = (-1000, 1)$. In this environment, the classifier which uses $\textrm{sign}(z_e)$ will have practically 100% test error, i.e. it will completely fail to generalize. To prevent this, our objective is to recover *only features whose informativeness of the label is invariant across all environments* (including those unseen). Thus, **calling the feature $\textrm{sign}(z_e)$ “invariant” is incorrect**, because this feature can vary across environments, up to and including reversing its correlation with the label. For your proposed feature, the optimal linear classifier is different in each environment, and therefore IRM *would not use it*; instead, IRM would learn the invariant classifier which always predicts the marginal $p(y)$—this is indeed minimax-optimal. **Furthermore, even if this feature *were* invariant, we note that it is non-linear, so it cannot possibly be a counter-example to Theorem 5.1.**
> >
> > We hope that this adequately demonstrates the distinction between the Bayes classifier and the invariant classifier and makes clear why the invariant classifier is indeed the objective of IRM. If this is still unclear, we further discuss the motivation for IRM in the section below on causality. We can now address specific questions which are relevant to the discussion above. Note that we have updated the submission; we will mirror your indexing for each remark, but the precise location of each of these notes is likely to have shifted.
> >
> > p.4 Section 4 §2: The claim that such a function will use only invariant features is in fact the *entire premise and motivation of IRM*. Theorem 9 of (Arjovsky et al.) demonstrates that under appropriate conditions, any featurizer whose optimal classifier is invariant will have the same optimal classifier for all possible test environments. Formally, this is saying that $E[y \mid \Phi(x)]$ is the same for *every possible environment*. In our setting, this is equivalent to invariance of $p(y=1 \mid \Phi(x)) = \sigma(\hat\beta^T \Phi(x))$. As we mentioned above, if $\hat\beta$ includes coefficients for the environmental features, it cannot be simultaneously optimal for each environment. Thus the intended solution, using only invariant features $z_c$, is instead $\Phi(x) = [z_c],\ \hat\beta = [\beta_c, \beta_0]$ (the invariant classifier).
> >
> > p.5 Theorem 5: We believe this question further indicates your confusion regarding the motivation for IRM, rather than a problem with our results. Theorem 5.1 demonstrates that for a given number of training environments, IRM will either succeed or fail in recovering the invariant classifier, which has minimax-optimal risk across all possible test environments. There is no discussion here of the “number” of test environments because the intent is to perform well in *any possible environment*, as opposed to performing well on average (which is the goal of ERM). This point is absolutely crucial; **if you are still unclear on why we do not mention the number of test environments, please let us know so we can discuss this further**.
> >
> > p.5 Corollary 5.2: Hopefully this is more clear in light of the above discussion. Same with your note for the conclusion.

---

> > > ### Author Response · Authors · 2020-11-14
> > > **Thanks for reading so thoroughly, but you've misunderstood IRM (3/3)**
> > >
> > > p.6 Theorem 5.3: This theorem, like the others in this work, is in reference to the model we present in Section 3. And yes, “feasible” means “satisfies the constraints of the optimization objective of IRM”.
> > >
> > > ---
> > >
> > > We understand your complaint regarding the use of the word “causal”, but the intent of this term was to informally relate our results to existing work such as (Peters et al. 2016) and give meaningful intuition for our model (the IRM paper does precisely the same thing, only intuitively and without presenting a full data model). There is indeed an important distinction in the framework of do-calculus, but we intentionally do not make this distinction. Our use of terminology from causality is restricted to the introduction and informal motivation of the model, purely for intuition purposes—as you noted, none of our theoretical results rely upon causality. That is, while a causal model is the inspiration for IRM and this work, an actual causal interpretation is not necessary; we will update the earlier sections to clarify that the causality terminology is not intended to be formal.
> > >
> > > The discussion of causality serves as the motivation for IRM, but this appears to be another major source of confusion in your review. Consider the example given in the IRM paper: classifying camels and cows. Both the animal features (humps, black and white spots) and the background features (sand, grass) are informative of the label. However, we want our classifier to correctly identify a cow, even in the desert. Therefore, IRM hopes to derive a featurizer which extracts only the features which are *semantically relevant* to the prediction task at hand, while ignoring features which are incidental and may or may not generalize. Here we use the term “relevant”, but we could also say “causal”: when a human looks at an image of a camel, which features of the image are *causal with respect to how that human labels the image?* With this context, **the claim that “IRM does not seek to use invariant features only, but to learn those invariant features, possibly by using non-invariant ones” is incorrect.** This would imply that IRM hopes to learn a transformation of the “sand/grass” feature which is invariant; from this discussion it is clear that it instead hopes to learn to *ignore this feature altogether, just as a human would*, because the “sand/grass” feature does not “cause” the image to depict a cow or a camel. If, as you suggest, IRM were to learn a function of “sand/grass” which is invariant in the training distributions, such a feature would not generalize to all future test distributions. **In fact, Theorem 5.1 says that such a non-generalizing function *cannot exist* in the linear case so long as $E > d_e$, but is *guaranteed to exist and be preferable under the IRM objective* otherwise; this is one of the clearly meaningful results of this work.**
> > >
> > > You’ve also written that “the so-called "invariant classifier", which uses only invariant features, is not the optimal invariant predictor sought by IRM.” **On the contrary, our theoretical results unequivocally prove that when $E > d_e$, the invariant classifier *is the global minimum of the IRM objective***. This result further validates the careful alignment between our formal data model and the informal motivation provided for the IRM objective.
> > >
> > > Thus, our informal discussion of causality—which mirrors that of (Arjovsky et al.)—is meant to further elucidate the motivation: IRM hopes to recover only features whose informativeness of the label do not vary. In relation to (Peters et al. 2016), these would be precisely the direct parents of the target variable, hence the use of the term “causal”.
> > >
> > > *Once again, we hope that with your continued feedback, we can reword the introduction/model discussion to prevent this misconception for future readers.*

---

> > > > ### Author Response · Authors · 2020-11-14
> > > > **Now to address your more detailed points**
> > > >
> > > > Addressing your more detailed notes:
> > > >
> > > > p.1 §3: These points feel rather pedantic and uncharitable. Yes, “non-causal” was meant to refer to features which are non-causal with respect to the target variable; we believe this is clear enough from context, and furthermore this is merely the introduction, so we didn’t aim for full formality. The same applies to your notes on the informal discussion in p.3 §5. We will rewrite to clarify that these discussions are meant to be intuitive rather than formal so as to be more approachable.
> > > >
> > > > p.2 §6: You are correct that IRM does not make this assumption; the IRM paper also does not have formal results that only depend on the data model. As we mentioned, one of the goals of our work is to formalize some of the purely intuitive arguments in Arjovsky et al. We do not believe this assumption is particularly strong (it is common to the works whose models we generalize), and further, *this assumption only serves to make invariant feature learning and prediction easier* (see constraint 2 in [1] for a similar argument, which, incidentally, also uses the term “invariant feature” in exactly the same way). Therefore, we would expect IRM to have an *easier time* with this setting, and yet we still demonstrate that it fails.
> > > >
> > > > p.3 §8: The conditional is implied by the joint, but *the joint is not implied by the conditional*. Our intent was to allow for either, and therefore including both is necessary. It just so happens that in our model the joint is fixed, but this may not be the case in all such models.
> > > >
> > > > p.4 Section 4: This was an intentional structure decision. We initially had Section 4 before Section 3, but the paper took too long to get to the point. We therefore reordered to present the minimum possible setup before giving an intuitive idea of our results. Since the actual objective of IRM is not necessary for this (only a discussion of its motivation/overall goal), we pushed that to the next section.
> > > >
> > > > p.5 Appendix C.1: No, the y-axis is meant to read “accuracy”. $\eta$ is a hyperparameter of the model and therefore it would not make sense to plot an independent variable on what is typically reserved for the dependent axis. The ERM performance is a flat line because it did not change with increasing environments. This is due to the fact that the environmental parameters were drawn from a non-zero prior, inducing an environmental bias in order to more closely match real-world data (perhaps ERM would start to perform well if we drew exponentially many environments). We apologize for not describing the setup in adequate detail; we will correct this in the next version.
> > > >
> > > > p.6 Figure C.2: This is given in the paragraph above Figure C.2. The dotted line represents $\sqrt{d_c}$; if the invariant parameters are drawn from a Gaussian prior, this is the expected “crossover” point, where the results of Theorem 5.2 cease to apply. This is intended purely as an intuition for how often such a result holds.
> > > >
> > > > [1] Understanding the Failure Modes of Out-of-Distribution Generalization. Nagarajan et al., 2020. https://arxiv.org/pdf/2010.15775.pdf

---

> > > > > ### Comment · AnonReviewer1 · 2020-11-17
> > > > > **Answer (3/3)**
> > > > >
> > > > > I would like to emphasize that I am not an advocate of IRM. I believe the IRM objective itself is questionable, and I am glad the authors are trying to showcase its limitations theoretically. I am also very respectful of the work the authors seem to have put into that paper, however I do not feel comfortable recommending acceptance of a theoretical paper which I perceive is fundamentally ambiguous. I wish my efforts and comments will help you improve the quality of your paper, so that as a result I might change my recommendation.
> > > > >
> > > > > I believe I do understand IRM, thank you, and I hope this discussing will help you comprehend why I have difficulty understanding your paper.
> > > > >
> > > > > PS: I wrote "the objective in OOD generalization is to find the Bayes classifier, not the "invariant classifier"." What I meant was the general OOD Bayes classifier,
> > > > >
> > > > > $$
> > > > > \min_h \mathbb{E}_{e,x,y}\left[L(y, h(x))\right] \text{,}
> > > > > $$
> > > > >
> > > > > of which IRM can be considered a regularized variant when $p(e)$ is uniform. Forgive me if I was not clear enough. I hope you will agree that you also use the term "Bayes classifier" in the paper without a proper definition. I therefore had to make a guess, which was the OOD Bayes classifier.
> > > > >
> > > > > Detailed comments:
> > > > > p.4 l.3: Observe that the invariant classifier is distinct from the Bayes classifier. -> Which Bayes classifier are you referring to ? The OOD Bayes classifier, which does not have knowledge of the environment ?
> > > > >
> > > > > $$
> > > > > \min_h \mathbb{E}_{e,x,y}\left[L(y, h(x))\right] \text{,}
> > > > > $$
> > > > >
> > > > > Or the Bayes classifier of each environment, which is
> > > > >
> > > > > $$
> > > > > \min_h \mathbb{E}_{e,x,y}\left[L(y, h(x,e))\right] \text{?}
> > > > > $$

---

> > > > > ### Comment · AnonReviewer1 · 2020-11-17
> > > > > **Answer (2/3)**
> > > > >
> > > > > The IRM paper defines what is an invariant predictor (Definition 3). This does seem to match somehow your own Definition 1, except that along the way you seem to wrongfully consider that this predictor is unique. You say "a true set" of invariant features, and "the" invariant classifier. But there may exist several sets of invariant features, say $(X_1,X_2)$ is a set of invariant features, such that $X_1,X_2,Y \perp E$. Then also $(X_1)$, $(X_2)$, and even $\emptyset$ are sets of invariant features. Each one of those sets of invariant features then has its own optimal, environment-independent predictor. Which one is then "the" invariant predictor ? The definition is incomplete, and therefore ambiguous.
> > > > >
> > > > > In the text, you say "we hope to extract and regress on invariant features while ignoring environmental features, such that our classifier generalizes to all unseen environments regardless of their parameters. In other words, the focus is on minimizing risk for the worst-case environment. We refer to the classifier which will minimize worst-case risk under arbitrary distribution shift as the invariant classifier". But what do you mean by environmental features ? If you hope to extract invariant features, then the empty set $\emptyset$ is an invariant feature, and your classifier will generalize to all unseen environments since $Y \perp E | \emptyset$. What are those environmental features you want to ignore ? It seems like you imply that those are the $z_e$ variables from your toy problem. Why ? What makes them environmental features ? How do you characterize an environmental feature, outside of your toy problem ? Consider the following example: I want to predict if a cup of water is salted. I can only measure the salt concentration in the water, $x$, which let's assume can vary between 0 and 1. The distribution of $y\in\{0,1\}$ is $p(y=1|x\geq 0.5)=1$, and $p(y=1|x<0.5)=0$, so that the optimal classifier is $h(x)=[x\geq 0.5]$. I can observe data from $E$ environments $e \in \mathcal{E}_{all}$, drawn from a uniform distribution, which all share the same label conditional distribution, $p(y|e)=p(y)>0$. However, the feature distribution is $p(x|y,e)=\delta(\mu_e+0.5y)$, with $\delta$ the Dirac distribution, and $\mu_e \in [0, 0.5[$ an environment dependent parameter such that $p(\mu_e)$ is uniform as well. Clearly we have that $x$ is not an environment-invariant feature. Does it make it an environmental feature ? If so, then what is the optimal invariant classifier in that case ? The one that relies only upon $p(y|\emptyset)$ ? There clearly exists an invariant feature obtained from the correct thresholding $[x\geq 0.5]$, which gives an OOD-optimal classifier, however it relies on the non-invariant feature $x$. Would you consider that IRM fails in that case, when it recovers the optimal OOD classifier, but relies on environmental features ?
> > > > >
> > > > > You write that your data model is completely described by eqs (1)-(3). I do not agree. How are the environments distributed in your example ? How are the environment parameters chosen ? I can understand that maybe the environment distribution $p(e)$ is not that important since you are looking for the worst-case risk as an OOD objective, however the parameter distribution $p(\mu_e,\sigma_e)$ matters, and is missing from your model description. What if all $\mu_e$ are strictly positive ? Then clearly $z_e$, your environmental features, do carry some kind of invariant information about $y$, in all environments. This I tried to showcase by a counter-example in my initial review. Without that piece of information, what do your results state then ? How should they be interpreted ? Do they hold for any distribution $p(\mu_e,\sigma_e)$ ? I believe not, as I have just showed.
> > > > >
> > > > > I note that the other reviewers are praising the intuitivity of the paper. I do agree that the paper feels intuitive to read, however I still believe that it carries ambiguity on several important points. After I now have re-read the paper several times, and I have tried to write things down by myself, I think the results presented in the paper might be answering a relevant question about IRM. However the fact that this required me such an effort to overcome the overall ambiguity of the paper, indicates that there might be a problem. I therefore urge the authors to make the very question they are trying to answer clearer and well-posed, in the general case, to alleviate the ambiguity regarding their "invariant classifier", and whether the question they ask is "does IRM recover the invariant classifier from our toy problem" or "does IRM fail at OOD generalization". I believe all of my concerns are a consequence of the fact that the central question the paper tries to tackle is not well-posed.

---

> > > > > ### Comment · AnonReviewer1 · 2020-11-17
> > > > > **Answer (1/3)**
> > > > >
> > > > > I thank the authors for their answer.
> > > > >
> > > > > First, let me comment back on causality. I am glad we agree that an actual causal interpretation of IRM is not necessary, and that none of your theoretical results rely upon causality. Let me add that the very concept of causality itself can be tricky, and can introduce a lot of questions and ambiguity, which I believe is not beneficial to your paper. As such I would recommend that you either do not discuss causality at all in the paper, or that you are more careful with the way you introduce and discuss causal concepts. In the revised manuscript, I still read "each environment corresponds to interventions on the SEM (Pearl, 2009) on just the non-causal mechanisms". Sorry if my comment felt pedantic and uncharitable, but I repeat that this is ambiguous, as every mechanism in a causal SEM is by definition causal. Please note that this causality thing is not necessarily part of my decision assessment, but rather a honest feedback with the aim to improve the paper. I must say that I, on my side, feel a bit sad to invest time and effort that I could use on my own research in order to provide a valuable feedback in the form of respectful and detailed comments, and to be called uncharitable. Here are some supporting arguments to help you understand my point about causality. You say "when a human looks at an image of a camel, which features of the image are causal with respect to how that human labels the image", and "the 'sand/grass' feature does not 'cause' the image to depict a cow or a camel". You seem to imply a causal system here, which is at the very least debatable. Maybe for collecting your image dataset, people were asked specifically to take pictures of cows. Then it is the concept of a cow which is a cause of the image pixels. Do you see the ambiguity here ? Another argument is the following. What if you apply IRM to a dataset that has no causal interpretation ? Well, I believe your derivations still hold. So why talking about causality and causal features to begin with ? Again, causality is a very slippery terrain, and it calls for caution. I understand you want to relate to the existing literature which takes inspiration from causality, however if you must do so I suggest you do that properly.
> > > > >
> > > > > Second, about the general ambiguity in the paper, which is I think my biggest concern. I believe the question your try to tackle in the paper is not well-posed, and as such your results do not really support your claim. In fact, I believe your results can be interpreted to support the opposite claim, when you say IRM fails because it does not recover the "invariant classifier", one could argue that IRM actually succeeds. Let me explain. In the IRM paper the goal is OOD generalization (from p.2 Contributions), supposedly achieved via the IRM objective, which is given a causal interpretation in terms of interventions. In your paper, I believe you question the OOD generalization ability of IRM, however your also mention that it does not recover the invariant classifier. Is the invariant classifier the one achieving the best OOD generalization ? The paper is not clear about that. From the abstract, I read that you aim at providing an "analysis of classification under the IRM objective", and provide conditions "under which the optimal solution succeeds or, more often, fails to recover the optimal invariant predictor". This is I believe the very source of the confusion. By "classification under the IRM objective", I understand you mean you will evaluate the capacity of IRM for OOD generalization. Following this train of thought, by "optimal invariant predictor", I tend to understand the optimal OOD predictor, and that you will show IRM fails to recover this predictor. But to this moment I am still confused what you mean by "optimal invariant predictor". This is the main question I asked you to clarify in my review.
> > > > >
> > > > > > I would appreciate if the authors could formally describe what they mean by "invariant classifier", and point me to the evidence where the IRM authors argue that IRM will recover such a classifier.
> > > > >
> > > > > I note that you have not answered that question. What do you mean by "invariant classifier" ?

---

> ### Author Response · Authors · 2020-11-19
> **Thanks for your continued feedback!**
>
> Thank you for taking the time to respond in such detail! We very much appreciate your comments; we apologize if that wasn’t clear in our rebuttal. We’ve made multiple updates to the paper to address your concerns; hopefully this demonstrates that we value your input. Through our prior exchanges, we see two themes: the formality of Def 1 and the discussion of causality/motivation for IRM. We will do our best to succinctly address these points.
>
> ---
> ## Definition of the Invariant Predictor
> The invariant predictor is *model-dependent* (i.e. it’s tied to eqs. 1-3). **We’ve updated Def 1 and added more discussion to make this clearer and more formal. We’ve also slightly modified our terminology to be consistent with Arjovsky et al: we now use “predictor” to refer to the composition $\beta \circ \Phi$ and “classifier” for the regression vector $\beta$. We hope this resolves your concerns with the definition.**
>
> The *optimal invariant predictor*—we added ”optimal” per your point that intuitively, even the empty-set is invariant—is the predictor defined by the composition of
> $$\Phi(x) = [z_c],\quad \beta = [\beta_c, \beta_0].$$
> The individual $\Phi$ and $\beta$ are not unique, but the resulting composition is.
>
> Note that $\Phi$ recovers $z_c$, which comprise *all* invariant features. Thus it is optimal in a **minimax sense** over the environments, as the parameters $\mu_e, \sigma_e$ can change **in a worst-case fashion**. This addresses another point: you claim eqs 1-3 don’t fully specify our model because we don't specify a distribution over $\mu_e, \sigma_e$; **we specifically allow them to change *arbitrarily*** rather than assuming a distribution. Providing such a guarantee is intentional: the goal of IRM is to avoid relying on “non-invariant” features which could shift arbitrarily at test time. You ask whether we're asking "does IRM recover the optimal invariant predictor" or "does IRM fail at OOD generalization". In our setting, **these questions are the same**: if $\Phi$ depends at all on $z_e$, then under the worst-case environment, the error can be arbitrarily high.
>
> Note that our Def 1 closely resembles Def 3 in Arjovsky et al: under the featurizer $\Phi$, the classifier $\beta$ is Bayes-optimal for *every environment*. This is because $\Phi$ recovers only features whose joint distribution with the label is invariant, and therefore $\mathbb{E}[y | \Phi(x)] = \sigma(\beta^T \Phi(x))$ in every environment. **This mirrors IRM, which expects the optimal classifier to be simultaneously optimal for every environment**—except they don’t provide a data model, so there’s no way to formally compare our model to their intuition.
>
> ---
> ## Causality/Motivation of IRM
> Existing work on IRM is motivated—at least intuitively—by a causal model of data, with little theory. The intent of this work was to demonstrate the failure of IRM *within a formal model which matches this intuition*. We do not address if or when this intuition applies in practice; we simply develop a model which faithfully formalizes the ideas behind IRM and demonstrate that the objective does not achieve its stated goals.
>
> **Your concerns with our discussion on causality and the intuition for IRM appear more focused on IRM itself, rather than our work in particular.** The present work is not a comment on the “believability” of this intuition, so we don’t formally address the implications. Instead, this work presents—*for the first time*—a formal codification of the IRM intuitions and demonstrates that the objective does not behave accordingly. We agree that existing work is not sufficiently formal, which is why we developed this work as a first step towards formalizing the intuition via a well-defined data model. **Please take into consideration that *any discussion* on prior work will necessarily use informal language, as the prior work we reference *is itself informal.***
>
> You describe our model as “toy”; we do not feel it is unrealistic, especially with non-linear $f$. Moreover, our results show that **even in this simple model with a built-in ground truth**, the number of environments IRM needs is prohibitively large. Note also that this model generalizes several previous works with similar analysis.
>
> Lastly, there still appears to be a **substantial misunderstanding of IRM**. You state “IRM can be considered a regularized variant [of the OOD Bayes classifier] when $p(e)$ is uniform”; we simply do not agree. While IRM can be viewed as minimizing the error over the *seen* environments with some kind of complicated regularization, it doesn’t assume a distribution over environments (we don’t even understand what “uniform” means; uniform over *what*?).
>
> To be clear, we do not claim IRM is the “right” way to approach OOD generalization. We wish only to show that, in a formal model which precisely encodes the intuitions, IRM does not accomplish its stated objective. **This is exactly what our results demonstrate.**
>
> Thanks for your continued feedback!

---

> > ### Comment · AnonReviewer1 · 2020-11-19
> > **Answer**
> >
> > Thank you for your response.
> >
> > I think at this point you've answered most of my concerns. If indeed the questions "does IRM recover the optimal invariant predictor" and "does IRM fail at OOD generalization" are the same for your toy problem, and you make that clear in the paper, then I believe this adresses my main concern.
> >
> > Note that I didn't mean anything negative by using the term "toy problem". I am convinced studying toy problems is a very healthy thing to do, and I have no concern regarding whether your setup is realistic or not. Showing theoretical failure in a simple scenario is sufficient to disprove the general success of a method. Also you mention that this setup generalizes previous works, so all for the best.
> >
> > I do agree that prior work on this topic is essentially informal and relies a lot on debatable causal intuition. Thank you for acknowledging that. Maybe making things right on that matter would require a separate paper. For the benefit of the community, I do suggest that you do your best to not repeat or encourage the same fallacies in your paper, if indeed you agree with me on that point.
> >
> > Regarding "IRM can be considered a regularized variant [of the OOD Bayes classifier] when $p(e)$ is uniform". I believe I was mistaken, you do not even need $p(e)$ uniform. Let's take a probabilistic perspective, and consider that training environments $e\in\mathcal{E}$ are i.i.d. samples from a probability distribution $p(e)$, defined over a potentially larger support $\mathcal{E}_{all}$, which is the support of the OOD minimax objective. Under this view, the ERM objective yields the OOD Bayes classifier (not the minimax OOD one),
> >
> > $$
> > h^\star=\arg \min_h \mathbb{E}_{e,x,y}\left[L(y, h(x))\right] \text{,}
> > $$
> >
> > which can be estimated empirically from the training environments as,
> >
> > $$
> > h^\star=\arg \min_h \sum_{e \in \mathcal{E} } \mathbb{E}_{x,y}\left[L(y, h(x))\right] \text{.}
> > $$
> >
> > This is precisely the IRM objective, from the original paper page 5, when we remove the invariance constraint. As such, I believe IRM can be considered a regularized variant of the ERM objective. An interesting question would then be, is the IRM objective biased ? Does IRM recover the OOD Bayes classifier, if it has access to infinite $(e,x,y)$ samples ? Another interesting question would be, does IRM recover the minimax OOD classifier, if it has access to infinite $(e,x,y)$ samples ? Of course to answer these questions one has to consider the environment distribution $p(e)$, on which the OOD Bayes classifier depends. But maybe the IRM classifier does not depend on $p(e)$ (if $p(e)>0$) ? That is another interesting question.
> >
> > I thank again the authors for their answer to my concerns. At this point I will wait for the discussions with the other reviewers in order to give my final recommendation.

---

> > > ### Author Response · Authors · 2020-11-19
> > > **Glad to hear we've addressed your concerns!**
> > >
> > > We’re very glad to hear we’ve reached the crux of your concerns and that we’re in agreement on so many points! We’d like to take one more opportunity to clarify the difference between IRM and “regularized ERM”. You appear to have the following understanding of the setup/goal of IRM:
> > >
> > > There is some distribution over environments $p(e)$, and we draw some number of environments $E$ and some number of samples from each of these environments. Then the goal is to minimize the population loss averaged over *all environments* (weighted by $p(e)$). The usual way of doing this is standard ERM (combining the samples from the observed environments), and you seem to view the IRM constraint as a regularizer which would somehow improve performance on future test distributions.
> > >
> > > In contrast to this “average-case” notion of OOD generalization, which requires we fully specify the distribution over environments—this is presumably why you felt our model was not “fully specified”—the notion considered by IRM is **minimax**, in that it hopes to perform well even in the **worst-case** environment. Thus, in actuality, IRM *does not assume any distribution over environments whatsoever*, and could in fact be expected to lead to *worse average performance than the ERM solution*. The rationale is that there may be  features which will correlate with the label across *a typical environment* (e.g., “grass” would correlate with “cow” in any real-world dataset). However, IRM wants to avoid using this feature because *theoretically, in the most adversarial case, there could be a test set which only depicts cows in the desert*. Whether you find this goal reasonable or not, this is the intent of IRM. We **highly recommend the paper which inspired IRM [1]** to see a more formal discussion on this topic.
> > >
> > > [1] Causal inference using invariant prediction: identification and confidence intervals. Peters et al. 2016

---

### Official Review · AnonReviewer2 · 2020-10-26
**Clear theoretical analysis.**

**Rating:** 7
**Confidence:** 2

**Review:**

Main comments:

This paper studies a theoretical aspect of IRM and how will it fail. Main contribution is pointing out that IRM is ineffective when the number of environments $E$ is smaller than the dimension of environmental feature $d_e$. A simple but universal model assumption is built, where environmental feature $z_e$ and causal feature $z_c$ is sampled from Gaussian conditional on label $y$. The analysis is two-fold: linear regime and non-linear regime. In the former part, given the feature extractor $\Phi$ is linear, a constructed solution to IRM is built to demonstrate the result. For the latter part, the other show the failure of IRM via several results in Thm 6.1 / D.3. The whole analysis is clear and easy to follow, thus the reviewer believe this submission deserves to be accepted.

Main comments:

The simulated experiments in Appendix C demonstrate the theoretical results. However, it's a toy run and  thus one drawback to the reviewer is that, for Colored MNIST experiments in IRM, it's obvious that $d_e > E$ while IRM still works. It would be great if the authors could give some explanation on this point.

Analysis in the paper is more a constructed one, whereas lacks of how optimization algorithm can lead to such a solution, which is another drawback of this paper.

Another minor comment:

One seminal reference seems missing: Causal inference using invariant prediction: identification and confidence intervals. Jonas Peters, Peter Bühlmann, Nicolai Meinshausen.

---

> ### Author Response · Authors · 2020-11-14
> **Thanks for your review**
>
> Thanks for your comments! To address your minor point, we note that we do cite this work more than once in the paper, including in the abstract. (Peters et al. 2016) was absolutely crucial to the development of this line of work (which we call “Invariant Causal Prediction”), so we very much agree that it deserves to be cited.
>
> To your larger concerns:
>
> * We acknowledge the limited setting for the experiments in Appendix C. However, we note that running IRM-related experiments in the non-linear regime is quite difficult and unstable; see our comment to Reviewer 1. Actually, there is an entire paper [1] based on running these experiments for comparisons.
> * The aforementioned reference [1] also addresses your other point: **it is not actually clear that IRM works on Colored MNIST**. In tandem with this theoretical work, several empirical works have recently demonstrated that performance of IRM and related objectives are extremely sensitive to training data, hyperparameters, etc. Further, there is increasing evidence that *none of these works outperform ERM* when it is tuned properly (again, see [1]). In particular, existing reports of performance of IRM on CMNIST *use the test set as a validation set*, which violates standard testing principles by leaking information. Thus, we challenge the notion that IRM works, even empirically, when $d_e > E$.
> * We agree that analysis of the optimization trajectory would be ideal. But that is a much more difficult question and one which warrants an entirely separate paper. The intent of this work was merely to demonstrate that even if we could solve the proposed objective, the resulting classifier would not behave as expected. If the optimum of an objective does not behave as we intend, this raises questions about the validity of the objective itself, with or without an analysis of the method by which the optimum is reached. In other words, this work demonstrates that an analysis of the trajectory is not even necessary, because we are optimizing the wrong objective to begin with.
>
> Please tell us if you have any other questions or concerns! In particular, let us know if there is anything we can clarify which would increase your confidence in the quality of this work.
>
> [1] In Search of Lost Domain Generalization. Ishaan Gulrajani and David Lopez-Paz, 2020. https://arxiv.org/abs/2007.01434

---

> > ### Comment · AnonReviewer2 · 2020-11-14
> > **The experiments in [1] are in different settings.**
> >
> > Thanks for your reply for my concerns. In [1] their settings are different:
> >
> > Let's denote the two train domains and one test domain in IRM paper as A, B, C. The IRM is designed to train on A, B and test on C. This is because the spurious feature is more correlated with label than causal feature on A, B, while is opposite on C. If causal feature is more correlated to label than spurious feature (which is the thing on C), then it's natural for ERM to beat IRM. In [1], the authors average the cases where A, B, C are taken as test domain independently. In this case it's natural to see IRM cannot beat ERM.

---

> > > ### Author Response · Authors · 2020-11-14
> > > **This is a good point, but the failure of IRM still stands**
> > >
> > > This is true! But it's still the case that there's very little evidence that IRM works better *even in the original setting*. As we pointed out, reported empirical performance relies on tuning on the test set. These experiments were intended to show only that IRM is *capable of learning invariance* with the correct tuning (such as with early stopping). In a more realistic setting, without access to the test set,  training with the IRM objective collapses to the ERM solution during training. In other words, we are not sure if there are any empirical results whereby IRM is trained in a fair manner, without access to the test set, and still manages to significantly outperform ERM.
> > >
> > > From [2]:
> > > "We emphasize, however, that [CMNIST] is methodologically
> > > unsuited for benchmarking, since, following Arjovsky et al.
> > > (2019) we assume access to the test distribution for hyperparameter tuning. Ultimately, these experiments should be
> > > interpreted only as a demonstration that, unlike ERM or RO,
> > > REX and IRM are both capable of OoD generalization in
> > > the face of spurious features, when properly tuned."
> > >
> > > [2] Out-of-Distribution Generalization via Risk Extrapolation. Krueger et al. 2020. https://arxiv.org/pdf/2003.00688.pdf

---

> > > > ### Comment · AnonReviewer2 · 2020-11-17
> > > >
> > > > My personal belief is that if there is a validation set which follows the same distribution with test set, then IRM can work. I do agree cmnist is more a showcase and unsuited for benchmarking.

---

### Official Review · AnonReviewer4 · 2020-10-28
**This paper takes a critical view of IRM. It shows under a particular DGP, in the linear case, a large number of environments are necessary for the recovery of the invariant predictor.  Furthermore, it shows that in the nonlinear case, IRM does not generalize to distributions that are different from the training environments.**

**Rating:** 7
**Confidence:** 2

**Review:**

- pros:
    - The paper examined the claims of IRM thoughtfully and critically.
    - the exposition is clear --- I particularly enjoyed the informal results in section 3.1. I applaud the authors' effort to make the results intuitive!
    - The appendix included illustrative and convincing empirical studies.
    - The paper brought up a failure case of IRM, which shows that there exists a nearly optimal classifier, similar to the invariant predictor in the training environment, but performs equal to the ERM solution in the testing set.
- cons
    - There is only one DGP used in throughout the paper.  In particular, the DGP  has the same complexity in the y->z_c and z_c -> y relationship. This seems counter-intuitive to me. I would be interested in learning if this phenomenon generalizes to other DGPs

---

> ### Author Response · Authors · 2020-11-14
> **Thanks for your review**
>
> Thank you for your review! We are glad to hear you appreciate the effort that went into stating the results intuitively; we took great care to present the results of this work both formally (with more in the appendix) and in plain English, to make it as accessible as possible.
>
> Addressing your concerns:
> * We assume DGP stands for “Data Generating Process”? If so, you are correct that we only used a single model in this work; we felt the model was flexible and general enough to justify this (e.g., this model generalizes [1], which also uses only one DGP).
> * We think it is quite likely that identical results could be demonstrated for the same graphical model but with Bernoulli-distributed features instead of Gaussian, but we didn’t feel that this would significantly contribute to the results, and it would have been difficult to fit in the page limit.
> * We actually view the mirrored $z_c \to y$ and $y \to z_c$ relationship as a *strength*, rather than a weakness. Many works on OOD generalization rely on the assumption that one or the other of these directions hold. By analyzing a model with both, we preempt future work from suggesting the other direction and claiming that our negative results do not apply.
>
> Please tell us if you have any other questions or concerns! In particular, let us know if there is anything we can clarify which would increase your confidence in the quality of this work.
>
> [1] An investigation of why overparameterization exacerbates spurious correlations. Sagawa et al., 2020. https://arxiv.org/pdf/2005.04345.pdf

---

### Official Review · AnonReviewer3 · 2020-10-30
**The work provides notable theoretical understandings on the invariant risk minimization scheme.**

**Rating:** 7
**Confidence:** 3

**Review:**

Pros:
* The work gives extended theoretical analysis on the effect of invariant risk minimization scheme, which is an increasingly popular framework for robust prediction. The work considerably extends the results in the original IRM paper. The results seem reasonable, and clarify implausible beliefs on the framework. The intuitions and proving techniques could also inspire new methods to improve the current framework.

Cons (minor issues):
* I know the main contribution of the work is on the theoretical side, but a simple experiment or simulation for the nonlinear case to demonstrate the empirical consequence would be appreciated.
* In the considered data generating model, it is assumed that $z_c \perp z_e | y$ (conditional independence). Is it required in the proofs? How natural/general can it be? For one example, it is mentioned that the causation can be viewed in the other direction, i.e. $z \to y$. In that case, observing $y$ would render components of $z$ correlated. Also, a graphical illustration of the data generating model would be helpful.
* OOD generalization to an arbitrary environment in the nonlinear case (even with the invariant optimal predictor on top of representations) may not be possible anyway, and some recent works on OOD generalization does not expect the learned model to work well in environments out of the convex hull/combination of training environments. So I am wondering how arbitrary can the test environment be in Theorem 6.1, e.g., can it be out of the convex hull/combination of training environment? The result would seem more interesting to me if the test environment can be.

=== EDIT: post-rebuttal ===

Thanks for the additional explanations.

---

> ### Author Response · Authors · 2020-11-14
> **Thank you for your review**
>
> Thank you for your review! To address your minor concerns:
>
> * Unfortunately, the lack of existing techniques for stable training, hyperparameter choice, etc. makes informative nonlinear experiments difficult (See the related discussion in [1]). We can point you to [2] for existing experimental evidence of the failure of IRM and other OOD generalization techniques in the non-linear regime.
> * Our proofs apply to the model we consider in the paper. While developing this work, we considered several models and achieved varying levels of success in showing similar results in each of them. It is not immediately clear to us if conditional independence is *necessary* to show such results, but our proofs in the linear case do use this fact; in the non-linear case we do not rely on such independence. We note that such an assumption would seem to make the job of learning invariances *strictly easier*, so the fact that IRM doesn’t work in this case is even more surprising. We agree that a graphical model depiction would be helpful; we intend to include this in the camera-ready version.
> * Theorem 6.1 makes no assumptions whatsoever on the test environment, other than being sufficiently far from the training environments. So it applies equally to test environments that are inside or outside the convex hull. In particular, this implies that even if some IRM-based technique *does* generalize to the convex hull of training environments in the linear case, it will not work in the non-linear case. This is because the “convex hull” argument typically relies upon restricting degrees of freedom and, as we discuss in section 6, this argument does not generalize when we introduce non-linearity.
>
> Please tell us if you have any other questions or concerns! In particular, let us know if there is anything we can clarify which would increase your confidence in the quality of this work.
>
> [1] Out-of-Distribution Generalization via Risk Extrapolation (REx). Krueger et al., 2020. https://arxiv.org/abs/2003.00688
> [2] In Search of Lost Domain Generalization. Ishaan Gulrajani and David Lopez-Paz, 2020. https://arxiv.org/abs/2007.01434

---

### Decision · Program_Chairs · 2021-01-07
**Final Decision**

**Decision:**

Accept (Poster)

**Comment:**

The authors have made significant efforts to thoroughly address all the concerns. Due to the amount of discussions, I had to go through the paper myself and agree with the authors on many of the points. In my opinion, this is a solid theoretical work on the pitfalls of IRM.